Manuscript prepared for Atmos. Chem. Phys.
with version 2014/09/16 7.15 Copernicus papers of the LaTeX class copernicus.cls.
Date: 28 October 2016

# Quantifying the Loss of Processed Natural Gas Within California's South Coast Air Basin Using Long-term Measurements of Ethane and Methane

Debra Wunch[1,2], Geoffrey C. Toon[2,3], Jacob K. Hedelius[2], Nicholas Vizenor[4], Coleen M. Roehl[2], Katherine M. Saad[2], Jean-François L. Blavier[2,3], Donald R. Blake[4], and Paul O. Wennberg[2,5]

[1]Department of Physics, University of Toronto, Canada
[2]Division of Geological and Planetary Sciences, California Institute of Technology, Pasadena, California, USA
[3]Jet Propulsion Laboratory, California Institute of Technology, Pasadena, California, USA
[4]Department of Chemistry, University of California, Irvine, California, USA
[5]Division of Engineering and Applied Science, California Institute of Technology, Pasadena, California, USA

*Correspondence to:* Debra Wunch (dwunch@atmosp.physics.utoronto.ca)

**Abstract.** Methane emissions inventories for California's South Coast Air Basin (SoCAB) have underestimated emissions from atmospheric measurements. To provide insight into the sources of the discrepancy, we analyse records of atmospheric trace gas total column abundances in the SoCAB starting in the late 1980s, to produce annual estimates of the ethane emissions from 1989–2015, and methane emissions from 2007–2015. The first decade of measurements shows a rapid decline in ethane emissions coincident with decreasing natural gas and crude oil production in the basin. Between 2010 and 2015, however, ethane emissions have grown gradually from about $13 \pm 5 \ \mathrm{Gg \cdot yr^{-1}}$ to about $23 \pm 3 \ \mathrm{Gg \cdot yr^{-1}}$, despite the steady production of natural gas and oil over that time period. The methane emissions record begins with one year of measurements in 2007 and continuous measurements from 2011–2016 and shows little trend over time, with an average emission rate of $413 \pm 86 \ \mathrm{Gg \cdot yr^{-1}}$. Since 2012, ethane to methane ratios in the natural gas withdrawn from a storage facility within the SoCAB have been increasing by $0.62 \pm 0.05\% \cdot \mathrm{yr^{-1}}$, consistent with the ratios measured in the delivered gas. Our atmospheric measurements also show an increase in these ratios, but with a slope of $0.36 \pm 0.08\% \cdot \mathrm{yr^{-1}}$, or $58 \pm 13\%$ of the slope calculated from the withdrawn gas. From this, we infer that more than half of the excess methane in the SoCAB between 2012–2015 is attributable to losses from the natural gas infrastructure.

# 1  Introduction

Anthropogenic sources of the potent greenhouse gas methane ($CH_4$) constitute about 60% of the global total $CH_4$ emissions, or nearly $350 \ TgCH_4 \cdot yr^{-1}$ (Saunois et al., 2016). Urban regions are thought to be an important contributor to this flux (e.g., McKain et al., 2012), and thus quantification and attribution of these urban sources are crucial for fully understanding their causes and hence potentially regulating them. Southern California's South Coast Air Basin (SoCAB) has been the focus of several studies. These studies have quantified the emissions from the basin and generally find that the SoCAB emissions are higher than the reported inventories (Wunch et al., 2009; Hsu et al., 2010; Townsend-Small et al., 2012; Wennberg et al., 2012; Peischl et al., 2013; Wong et al., 2015; Hopkins et al., 2016; Wong et al., 2016).

The SoCAB is a highly urbanized region centered on Los Angeles, with almost 17 million residents, representing 43% of the population of California. The lower atmosphere over the SoCAB is well-confined: it is contained by mountains to the north and east, and open to the Pacific Ocean to the south-west. Thus, urban emissions within the basin have long residence times and, under prevailing wind conditions, also have strong and predictable diurnal flow: out to the ocean at night, and inland during the day.

The many sources of methane in the SoCAB include oil and gas exploration and extraction, natural gas delivery pipelines and storage facilities, waste-water treatment plants, landfills, and dairies. Previous studies have shown that the atmosphere over the SoCAB contains significant $CH_4$ enhancements over the global background (Wunch et al., 2009; Hsu et al., 2010; Wong et al., 2015). More recent work has attempted to attribute the sources of the enhanced methane using other tracers in the atmosphere that are co-emitted with particular sources. Wennberg et al. (2012) used simultaneous measurements of ethane ($C_2H_6$) and methane to separate ethane-containing sources of methane, such as natural gas and petroleum, from biogenic sources of methane which do not co-emit ethane, such as landfills, waste water treatment and ruminants. Wennberg et al. inferred that a significant fraction of the excess methane in the SoCAB atmosphere is likely emitted from the natural gas infrastructure, potentially post-consumer meter. Peischl et al. (2013) used co-emitted higher-order alkanes (including ethane) to suggest that oil and gas drilling and storage are significant contributors to the elevated methane and ethane emissions. Hopkins et al. (2016) and Townsend-Small et al. (2012) conclude that most of the elevated methane in the western SoCAB is related to fossil fuels using spatial alkane measurements and isotope measurements, respectively.

We describe our data records and analysis methodology in §2, and, in §3, we discuss the change in the emissions of methane and ethane within the SoCAB. By comparing the ethane to methane ratios measured in the atmosphere with the changing ratios in the withdrawn and delivered natural gas, we quantify the fraction of the excess methane in the atmosphere attributable to the natural gas infrastructure.

## 2 Methods

We use data from four solar viewing ground-based Fourier transform spectrometers (FTS) that have measured within the SoCAB. The first instrument, the JPL MkIV FTS (Toon, 1991), has measured ethane, methane and other trace gases from the Jet Propulsion Laboratory (JPL, NASA) since 1985 (Figure 1). The measurements have been made once or twice per week, for about 2 hours per day, when the instrument is not in the field elsewhere for intensive scientific campaigns. Two other instruments were temporarily stationed at JPL: JPL2007 (Wennberg et al., 2014c; Wunch et al., 2009) was operational between July 2007 and June 2008, and JPL2011 (Wennberg et al., 2014a) was operational between July 2011 and July 2013. These instruments measured $CH_4$ and other gases, but not $C_2H_6$, and are part of the Total Carbon Column Observing Network (TCCON, Wunch et al., 2011). The fourth instrument, which is located about 10 km from JPL at the California Institute of Technology (Caltech), is part of the TCCON, and has been measuring ethane, methane and other trace gases with high temporal frequency (several hundred spectra per sunny day) since September 2012 (Wennberg et al., 2014b). The JPL MkIV FTS data are available from the MkIV website (http://mark4sun.jpl.nasa.gov/ground.html), and the TCCON data are available from the TCCON archive (http://tccon.ornl.gov/).

Both the MkIV and TCCON FTS instruments are direct solar-viewing and measure solar absorption by atmospheric trace gases; the retrievals are thus insensitive to atmospheric aerosol abundances. The data analysis for these instruments makes use of the GGG2014 software package (Wunch et al., 2015). This includes a nonlinear least squares spectral fitting algorithm (GFIT) that scales an *a priori* profile for best fit, and a spectroscopic linelist (Toon, 2014) based on the HITRAN database (Rothman et al., 2013). The GGG2014 software produces column-averaged dry-air mole fractions of the trace gas of interest ($X_{gas}$), which is defined as:

$$X_{gas} = \frac{\text{column}_{\text{gas}}}{\text{column}_{\text{air}}^{\text{dry}}} \tag{1}$$

The column of dry air, in units of $\text{molecules} \cdot \text{cm}^{-2}$, is computed either from retrieved oxygen ($O_2$) when available (for the TCCON records), or from precise measurements of the surface pressure (for the MkIV record):

$$\text{column}_{\text{air}}^{\text{dry}} = \frac{\text{column}_{O_2}}{0.2095} \tag{2}$$

$$= \frac{P_s}{\{g\}_{air} m_{air}^{dry}} - \text{column}_{H_2O} \frac{m_{H_2O}}{m_{air}^{dry}} \tag{3}$$

The measured surface pressure ($P_s$) is converted to a dry surface pressure by subtracting the column amount of water ($\text{column}_{H_2O}$), where $\{g\}_{air}$ is the column-averaged gravitational acceleration, $m_{air}^{dry}$ is the molecular mass of dry air, and $m_{H_2O}$ is the molecular mass of water.

The MkIV time series plots shown in Figure 1 reflect the influence of local sources in addition to the large scale backgrounds for these gases. To show the global background trends, overlaid on Figure 1 are the surface *in situ* measurements of methane (Dlugokencky et al., 2016), carbon monoxide

(CO, Novelli and Masarie, 2015), and ethane (Helmig et al., 2015) made atop Mauna Loa, Hawaii. The apparent "noise" in the MkIV time series is both from diurnal changes and from the larger sea-
90 sonal changes. Note that the magnitude of the Mauna Loa free-troposphere *in situ* concentrations should not be expected to exactly match the MkIV total column-averaged dry-air mole fractions. In particular, the concentration of methane is significantly lower above the tropopause, and so $X_{CH_4}$ is generally lower than the free-tropospheric methane concentrations (Washenfelder et al., 2003; Saad et al., 2014; Wang et al., 2014).

To diagnose the contribution of SoCAB sources to the trace gas columns, we quantify the diurnally-varying gas ratios following the methodology described in detail in Wunch et al. (2009), and briefly described as follows. Because of the topography of the SoCAB and its predictable diurnal wind flow pattern, gases emitted into the basin atmosphere, even if they are not emitted by the same source, show similar diurnal patterns, with a peak in the total column around 2 pm local time, when the
planetary boundary layer is thickest. Diurnal changes thus represent emissions into the SoCAB. To quantify the diurnal change in $X_{gas}$ for the TCCON data, we subtract morning values from after-noon values at the same solar zenith angles, producing $\Delta X_{gas}$, a "gas anomaly" value. This approach minimizes airmass-dependent biases in the measurements from appearing as diurnal changes, but it does not remove the small temperature bias (as afternoons are systematically warmer than morn-
ings). However, sensitivity studies which perturb the assumed lower atmosphere temperature show that the temperature bias has a small effect on the diurnal change of the trace gases, with magnitudes of $\lesssim 5\%$ of the total diurnal variability (see Appendix B).

We assume that the emissions into the lowest layers of the atmosphere cause the diurnal pattern in $X_{gas}$ and thus we explicitly account for differences in the measurement sensitivity at the surface to
110 each gas by dividing the $\Delta X_{gas}$ by the value of the column averaging kernel at the surface. We then compute the slope that relates anomalies of one gas to another. Our data filtering scheme, designed to minimize the impacts of non-basin air, fires, significant weather events, and instrument problems is described in Appendix A.

The MkIV dataset is temporally sparse, and the observation strategy was not intended for this
kind of differential analysis: MkIV measurements are taken around solar noon, and only for one to two hours per day. While this observation strategy minimises airmass variation, columns measured only an hour apart tend to be similar, and so the computed anomalies are small and therefore noisy. A consequence of this is that MkIV methane measurements, which have smaller fractional diurnal variability than the other gases presented here, are not currently precise enough for anomaly analysis.
Daily anomalies of ethane, carbon monoxide and acetylene are computed here by subtracting the daily mean value from each measurement, and applying the column averaging kernel in the same manner as for the TCCON datasets. We aggregate MkIV $\Delta X_{gas}$ data for each year to calculate tracer-tracer anomaly slopes. Because the TCCON datasets are much denser, we aggregate monthly

data. Subsampling the TCCON datasets to match the times of the MkIV measurements does not appear to bias the results (see Appendix C).

To determine emissions of the gas of interest, we use tracer-tracer anomaly slopes to carbon monoxide (CO), whose emissions in the SoCAB are well constrained by extensive, biannual, mandatory vehicle smog checks and oversight by the California Air Resources Board (CARB), and are published through the CARB webpage by air basin (http://www.arb.ca.gov/app/emsinv/emssumcat.php). Wunch et al. (2009) suggested that using CO instead of $CO_2$ to compute emissions may underestimate the emissions due to different diurnal emissions patterns, but subsequent studies have shown better agreement with the $CH_4$ emissions estimates computed using its relationship with CO (Wennberg et al., 2012; Peischl et al., 2013; Wong et al., 2016). To calculate the emissions of the gas of interest, we apply the following equation:

$$E_{gas}^{SoCAB} = \left( \alpha_{gas} \frac{M_{gas}}{M_{CO}} \right) E_{CO}^{SoCAB} \tag{4}$$

where $E_{CO}^{SoCAB}$ is the emission of carbon monoxide in the SoCAB in units of $\mathrm{TgCO}$, $\alpha_{gas}$ is the slope of the correlation between the gas of interest and carbon monoxide in $\mathrm{mol \cdot mol^{-1}}$, $M_{gas}$ and $M_{CO}$ are the molecular masses of the gas of interest and carbon monoxide, respectively, in $\mathrm{g \cdot mol^{-1}}$.

The uncertainty estimates on the tracer-tracer anomaly slopes are the standard deviation of many slopes calculated by bootstrapping (Efron and Gong, 1983) a linear fit that takes x- and y-errors into account (York et al., 2004). Uncertainty estimates on the emissions are determined by multiplying the calculated emissions by the sum in quadrature of the fractional uncertainties of the slopes and the assumed uncertainty on the CARB carbon monoxide emissions (20%).

## 2.1 Ancillary Data

To determine the composition of the natural gas delivered to the SoCAB, we collected bi-weekly samples of the natural gas delivered to Caltech by SoCalGas. Natural gas components were separated using gas chromatography on an HP-PLOT Q column. The abundance of each gas was measured using a flame ionization detector with appropriate calibrations. To ensure no drift in the chromatograph, a natural gas standard was also regularly analyzed. Prior to November 2014 the analysis was performed on site on the same day the sample was collected. Afterwards, samples were collected in canisters and analyzed in batches using an off-site gas chromatograph, also using a PLOT column and flame ionization detector.

To determine the composition of the natural gas stored within the SoCAB, we use data made publicly available by the Southern California Gas Company (SoCalGas). There are four SoCalGas gas storage facilities (Aliso Canyon in Northridge, Honor Rancho in Valencia, Golita near Santa Barbara, and Playa Del Rey), two of which are within the SoCAB (Aliso Canyon and Playa Del Rey). Both the Aliso Canyon and Playa Del Rey facilities are exhausted oil wells that were re-purposed to store natural gas. The Aliso Canyon facility is one of the largest depleted-well gas storage facilities

in the United States, with an 168 billion cubic foot capacity (4.8 billion cubic meters) (AQMD, 2016; USEIA, 2016); the Playa Del Rey facility can store only about 2 billion cubic feet (~1% of the Aliso Canyon capacity). As the result of a 2007 legal settlement, SoCalGas publishes monthly withdrawn gas composition from the Playa Del Rey wells (SoCalGas, 2008). The data are freely obtained from their website (https://www.socalgas.com/stay-safe/pipeline-and-storage-safety/playa-del-rey-storage-operations). The Aliso Canyon facility does not regularly make their withdrawn gas composition publicly available. However, between October 2015 and February 2016, they made daily atmospheric measurements near the facility available on their website in response to the failure of one of the withdrawal wells resulting in a large loss of gas (https://www.alisoupdates.com/acu-aliso-canyon-air-sample-results). Other measurements from aircraft near the facility have been recently published (Conley et al., 2016).

## 2.2   Defining local plumes within the data

Highly local plumes of methane are periodically observed throughout the Caltech FTS time record. We define these "plumes" as a diurnal change in methane that is not correlated with an associated change in carbon monoxide. Carbon monoxide is a heavily emitted gas within the SoCAB, but it has no significant common sources with methane, so correlations between carbon monoxide and methane are due to the SoCAB's atmospheric dynamics and thus represents what we will refer to as the "ambient" SoCAB air.

To quantify this, we use quantile-quantile plots (Wilk and Gnanadesikan, 1968) that determine whether two datasets draw from the same probability distribution. In these plots, a linear relationship indicates that the distributions are similar, and any deviations from linearity suggest that the distributions are different. We assume that the data in the linear region of the graph sample ambient SoCAB air, and the nonlinear regions are from the plumes. Figure 2 shows the quantile-quantile plots for anomalies in methane and carbon monoxide.

From these plots, we determine the regions of nonlinearity, marked by grey bars. We assume that the data that fall outside the grey bars represent air that is not well-mixed (i.e., "plume" air) and that the "ambient" air is contained in the box defined by the grey bars. The top panel shows the data prior to October 22, 2015 and after February 11, 2016, and the bottom panel shows the data between those dates, during the period of sustained Aliso Canyon losses.

## 3   Results and Discussion

We have computed emissions estimates of $C_2H_6$ since 1989 (Figures 3, 4), and $CH_4$ emissions estimates since 2007 (Figure 5). The emissions of ethane in the basin decreased significantly from the late-1980s (Figure 3) from $70 \pm 17 \, \mathrm{Gg \cdot yr^{-1}}$ to $13 \pm 5 \, \mathrm{Gg \cdot yr^{-1}}$ in 2010. These 2010 emissions values agree well with previous studies (12.9 Gg, Wennberg et al. (2012); 11.4±1.6 Gg, Peischl

et al. (2013)). Since 2010, however, ethane emissions have nearly doubled. Emissions of $CH_4$ are steady over the 2007–2016 period, with an average value of $413 \pm 86 \; Gg \cdot yr^{-1}$ and a slope of $-5 \pm 4 \; Gg \cdot yr^{-1}$ (-1.2±1.0 % $\cdot yr^{-1}$), in good agreement with the results from Wong et al. (2016), who have monitored $CH_4$ in various locations throughout the SoCAB since 2011.

There are three main sources of ethane emissions in the SoCAB: vehicle exhaust, the natural gas system, and oil and gas exploration and extraction. Of these sources, only vehicle exhaust is not a significant source of $CH_4$. To distinguish between vehicle exhaust and fossil fuel sources, we use our coincident measurements of carbon monoxide, which tracks sources of incomplete combustion (including mobile sources), and acetylene ($C_2H_2$), whose emissions more directly track vehicle exhaust (Kirchstetter et al., 1996; Warneke et al., 2012; Crounse et al., 2009). The ratio of ethane to carbon monoxide in the SoCAB declined rapidly until the mid-1990s, and then slowly and steadily increased. The ratio of acetylene to carbon monoxide remained relatively constant throughout the time period (Figures 3, 4), and thus the ethane to acetylene ratios follow the same trend as ethane to carbon monoxide. This implies that vehicle emissions are not driving the changes in ethane emissions. This is consistent with the Warneke et al. (2012) analysis, which showed an increase in ethane relative to acetylene after 1995, which they attributed to natural gas use and production. Using the motor vehicle gas composition measured by Kirchstetter et al. (1996), and the reported SoCAB carbon monoxide emissions for 1995 by CARB for mobile sources ($2.114 \; Tg \cdot yr^{-1}$, CARB, 2009), we infer that ethane emissions from mobile sources account for only ~8% of the observed ethane, in agreement with the $5 - 10\%$ estimate of Peischl et al. (2013) for the year 2010. Thus, emissions from vehicles are unlikely to be either a dominant source of ethane to the SoCAB atmosphere, or responsible for the significant decrease in ethane after 1995. Prior to 1995, there were fewer regulatory controls on air pollution from vehicles, and the exhaust composition is much less well-known (Kirchstetter et al., 1996).

Natural gas and crude oil production from the Los Angeles Basin decreased by about a factor of two between 1990 and 2000 (USEIA, 2015c, a). The region's natural gas liquids production, which includes ethane, propane and higher-order alkanes, is negligibly small and no production is reported after 1993 (USEIA, 2015b). The Los Angeles Basin and the SoCAB are not identical regions: the Los Angeles Basin encompasses the SoCAB except for the northwestern corner of Los Angeles County, but it additionally includes the eastern portions of San Bernardino and Riverside counties, and all of San Diego and Imperial counties. We assume that the production in the SoCAB tracks the Los Angeles Basin production. The fractional decrease in natural gas and crude oil production is consistent with the drop in ethane emissions measured by the MkIV FTS between 1990 and 2000 (Fig. 6). However, the absolute abundance is inconsistent with the 17% losses from oil and gas extraction determined by Peischl et al. (2013) for 2010: it would account for less than half of the $C_2H_6$ emissions in 1990. This suggests that either extraction losses from oil and gas production in the 1990s were significantly higher, or that the ethane content of the gas was larger.

Between 2000 and 2010, the ethane emissions remained relatively constant (Figure 3), consistent with the steady production of gas and oil. After 2010, however, the calculated ethane emissions increase monotonically, in contrast with the near-constant oil and gas production.

To explain the ethane increases in the latter period, we rely on our temporally denser atmospheric measurements from the Caltech FTS, combined with measurements of ethane and methane available

from the withdrawn natural gas composition of the Playa Del Rey storage facility, and measurements of the delivered natural gas composition to Caltech. Figure 7 shows the time series of ethane to methane ratios since late 2009 from the Playa Del Rey storage facility. The ratios were roughly constant at around 2.3% until a minimum in spring 2012 of ~1.7%. Since that time, the ethane to methane ratios have increased at a rate of $0.62 \pm 0.05\% \cdot \mathrm{yr}^{-1}$ with ratios exceeding 4% by mid-

2015. This significant increase in ethane content of the natural gas provides an unique opportunity to attribute the sources of $CH_4$ to the SoCAB atmosphere. Our measurements of the ethane to methane ratio in the natural gas delivered to Caltech show values consistent with the stored natural gas at Playa Del Rey and at Aliso Canyon and a consistent change in ratio over time ($0.59 \pm 0.10\% \cdot \mathrm{yr}^{-1}$). The variability of the ratios measured in the delivered gas is much higher than that reported by

SoCalGas (Figure 7) and commensurate with the variability seen in the atmospheric measurements. Since Caltech and Playa Del Rey are located ~45 $\mathrm{km}$ apart, this suggests that the Playa Del Rey withdrawn gas values provide a reasonable (if smoothed) approximation of the basin-wide natural gas ratios.

Measurements of the atmospheric ethane to methane emissions ratios using the Caltech FTS data

increase by $0.36 \pm 0.08\% \cdot \mathrm{yr}^{-1}$, which is 58±13% of the change in the ratio of ethane to methane reported in the storage gas by SoCalGas at the Playa Del Rey storage facility. The linear relationship between the Caltech FTS ethane to methane ratios and the Playa Del Rey ratios has a slope of $58 \pm 12\%$ (Figure 8), providing confirmation of this value. This finding is consistent with more than half of the excess atmospheric burden of methane in the western SoCAB being attributable to

emissions from the natural gas infrastructure.

Since the average total methane emissions in the SoCAB since 2007 have been roughly constant at $413 \pm 86 \ \mathrm{Gg} \cdot \mathrm{yr}^{-1}$ (Figure 5; Table 1), the ~58% attributable to the natural gas infrastructure is $240 \pm 78 \ \mathrm{Gg} \cdot \mathrm{yr}^{-1}$. In 2015, the SoCalGas total throughput was 2559 $\mathrm{MMcf} \cdot \mathrm{day}^{-1}$, or 18 $\mathrm{TgCH_4}$ total (California Gas and Electric Utilities, 2016). We remove 3 $\mathrm{TgCH_4}$ from wholesales,

and 0.2 $\mathrm{TgCH_4}$ for company use and "lost and unaccounted for" (LUAF) gas, giving 14.7 $\mathrm{TgCH_4}$ delivered by SoCalGas. This suggests 1.6±0.5% losses as fugitive emissions from the total delivered. (However, only 74% of the population served by SoCalGas lives in the SoCAB, and thus the fraction of the losses as fugitive emissions would represent a larger fraction of the delivered gas to SoCAB customers (Wennberg et al., 2012).) The roughly constant total $CH_4$ emissions and delivered natural

gas implies that downstream natural gas emissions were not likely changing during this period. The remaining ~173±56 $\mathrm{Gg} \cdot \mathrm{yr}^{-1}$ excess methane is likely from sources lacking an ethane signature

that tracks the pipeline natural gas composition. These likely sources are the SoCAB dairies (Viatte et al., 2016), feedlots and range cattle, landfills, septic systems (Wennberg et al., 2012), and – likely particularly important in the western part of the basin – oil and gas extraction. Peischl et al. (2013) estimate $182 \pm 54$ $GgCH_4 \cdot yr^{-1}$ emitted from methane-dominant sources (i.e., dairies, landfills and wastewater treatment plants), and the oil and gas extraction to be $32 \pm 7$ $GgCH_4 \cdot yr^{-1}$. Our results are consistent with these previous studies within the uncertainties. Table 1 compiles these emissions for $CH_4$ between 2007-2015 and for $C_2H_6$ for 2012-2015. We assume constant total emissions of $CH_4$ during the 2007-2015 period and changing $C_2H_6$ emissions from the increasing ethane content in the pipeline-quality natural gas. Within the uncertainties, the increase in observed $C_2H_6$ emissions can be wholly explained by the increasing ethane content in the delivered natural gas. The other sources of $C_2H_6$ (vehicular exhaust, oil and gas exploration and production) are assumed to be constant.

Droughts such as the one plaguing Southern California since 2012/2013 (Swain et al., 2014; Griffin and Anchukaitis, 2014) can reduce the ability of soil microbes to remove methane and ethane released underground into the soils (van den Pol-van Dasselaar et al., 1998; Adamse et al., 1972). The constant $CH_4$ emissions and growing $C_2H_6$ emissions since 2012 would require a compensating decrease in biogenic emissions of $CH_4$ to offset this effect. However, biogenic emissions are reported to have decreased by about 1% between 2012 and 2014 (CARB, 2016), so this effect is likely to be small.

### 3.1 Aliso Canyon

A large gas loss from the Aliso Canyon Storage Facility to the SoCAB began on October 23, 2015 according to SoCalGas and reports from those living nearby. The failed well was finally plugged on February 11, 2016. Conley et al. (2016) estimate that approximately 97.1 Gg $CH_4$ were released into the atmosphere during the 112-day leak, about 25% of the typical annual SoCAB methane emissions. After October 23, 2015, we see several days with very large enhancements in atmospheric methane and ethane, typically in the afternoons when the plume is advected into the line of sight of the instruments. We see no evidence of such large plumes prior to October 23 in our measurements. The plumes from Aliso Canyon can be easily distinguished from the ambient SoCAB air during this period (Figure 2, lower panel), and in these plumes, the ethane and methane anomalies are very well correlated with a slope of $4.28 \pm 0.07\%$ (Figure 9), in good agreement with the recent delivered natural gas ethane to methane ratios which exceed 4%. From our atmospheric measurements and the Conley et al. (2016) $CH_4$ emissions estimate, we calculate that the ethane emission from this leak is $7.7 \pm 1.7$ Gg $C_2H_6$, which is about 40% of the annual SoCAB ethane emissions. Conley et al. (2016) estimated a consistent 7.3 Gg $C_2H_6$ emissions using aircraft measurements.

While dramatic and important to prevent, the Aliso Canyon well failure represents only a small fraction of the SoCAB methane emissions over the long term ($<3\%$ of the emissions from the So-

CAB between 2007 and 2015). Furthermore, the annual methane emissions into the SoCAB ($10.3 \pm$ $2.2\,\mathrm{Tg\,CO_2e \cdot yr^{-1}}$, using the 100-year global warming potential of 25) represent less than 7% of those of carbon dioxide ($CO_2$), which we estimate to be $167.4\,\mathrm{Tg \cdot yr^{-1}}$ by scaling the California Air Resources Board estimate for California's carbon dioxide emissions in 2013 ($386.6\,\mathrm{Tg \cdot yr^{-1}}$, CARB, 2015) to the population of the SoCAB. Thus, significantly reducing the long-term climate impact of the SoCAB's greenhouse gas emissions requires focusing efforts to reduce carbon dioxide emissions directly.

## 4  Conclusions

We have measured the total column atmospheric abundances of ethane, methane and other trace gases since the late 1980s in the South Coast Air Basin in Southern California, USA. We calculate that ethane emissions declined rapidly until the mid-1990s, coincident with the decline in Los Angeles Basin production of natural gas and crude oil, but the absolute abundances are inconsistent with recent estimates of natural gas emissions from the SoCAB oil and gas production. This may suggest that either extraction losses were higher in the 1990s than they are today, or that the ethane content of the gas was larger. After the mid-1990s, the ethane emissions are relatively constant until ~2010, and then roughly double between 2010 and 2015. This increase cannot be explained by the (decreasing) vehicular emissions or (steady) natural gas and oil production in the basin, but can be explained by the increasing ethane content of the natural gas delivered to the SoCAB. Methane emissions have remained steady since 2007 at $413 \pm 86\,\mathrm{Gg \cdot yr^{-1}}$. Since 2012, ethane to methane ratios in the stored and delivered natural gas have increased, and are tracked in our atmospheric measurements with a slope of about $58 \pm 13\%$ the magnitude, implying that over half of the excess methane in the basin air is from losses in the natural gas infrastructure. These long-term measurements allow us to monitor the atmospheric composition and attribute changes in the atmosphere to specific sources within the basin with unique time dependencies.

The Aliso Canyon Gas Storage facility well failure on October 23, 2015, was one of the biggest singular natural gas releases in US history. Our measurements indicate that this leak, which is estimated by Conley et al. (2016) to have released $97.1\,\mathrm{Gg\,CH_4}$ into the SoCAB atmosphere in just 112 days, produced $7.7 \pm 1.7\,\mathrm{Gg\,C_2H_6}$, about 40% of the typical annual ethane emissions in the basin. The long-term climate impacts from the Aliso Canyon well failure are much smaller than the accumulated background methane emissions, and minor compared with the direct carbon dioxide emissions in the SoCAB.

## Appendix A:  Data Filtering

Data from the Caltech FTS ($N = 73335$) were filtered to avoid biases in the slopes using the following criteria:

– There must be at least 5 measurements during the day to calculate $\Delta X_{gas}$ anomalies.

– We filter out days on which the $\Delta X_{CO_2}$ changes by less than 1.5 ppm, as those are typically days during which the prevailing winds are so-called "Santa Anas," which bring relatively clean air from the Mohave Desert from the North into the SoCAB and hence are not representative of SoCAB air.

– We filter out days on which hydrogen fluoride anomalies ($\Delta X_{HF}$) change by more than 10 ppt. $\Delta X_{HF}$ is a proxy for tropopause height, and large changes in it over the course of the day indicates a front or other significant weather change not representative of typical SoCAB air.

– We filter out days on which the biomass burning tracer $\Delta X_{HCN}$ changes by more than 0.5 ppt, because these data are likely contaminated with fire emissions.

– Each month of data must contain at least 15 $\Delta X_{gas}$ points for a slope to be calculated for that month. This avoids biasing the slopes based on a few non-representative measurements.

– Ethane and methane are measured on two separate detectors: ethane is measured with an InSb detector; methane with an InGaAs detector. Both detectors measure carbon monoxide, and so we ensure that the carbon monoxide measured on the two detectors are consistent. Any measurements for which the carbon monoxide in the two bands differ by more than $2\sigma$ of their mean difference are excluded from further analysis.

Data from the MkIV FTS were filtered more loosely ($N = 1727$) than the Caltech FTS measurements, as the density of measurements is much lower, and measurements are manually initiated and terminated within a few hours of noon on clear, smoke-free days.

– There must be at least 5 $\Delta X_{gas}$ anomalies per year to calculate the tracer-tracer slopes.

– The change in $X_{CO}$ must be sufficiently large ($5 \times 10^{17}\,\text{molecules} \cdot \text{cm}^2$, or ~2%) in order to calculate a robust slope for each year.

**Appendix B: Temperature Bias**

The GGG2014 analysis software uses a single a priori temperature profile throughout each day that is representative of the local noon temperature profile, derived from the NCEP/NCAR reanalysis data (Wunch et al., 2015). On sunny days, there is a systematic increase in surface temperature throughout the day in the SoCAB; typically a 5 K difference between mid-morning and mid-afternoon at the surface (see Figs. B1 and B2); temperature changes aloft should be smaller and thus the integrated temperature error throughout the planetary boundary layer should be smaller than 5 K.

To minimize the temperature sensitivity of our retrievals, we chose windows in which the target absorption lines have average ground-state energies of around 300 $cm^{-1}$. For example, we use the entire CO and $CH_4$ bands in the near infrared, which have roughly the same number of high-j and low-j lines, reducing the temperature sensitivity. $C_2H_6$ is measured in its Q-branches between 2976 and 2997 $cm^{-1}$. Based on performing $C_2H_6$ retrievals using correct and incorrect (perturbed) temperature profiles under a range of different temperature and humidity conditions, we have determined that the retrieved $C_2H_6$ amount will change by less than 1% for a temperature perturbation of 5 K at the surface, decreasing to zero at 3.5 km altitude. Since a typical diurnal change between mid-afternoon and mid-morning in the retrieved $C_2H_6$ is about 20%, the temperature-induced affect is comparatively small. A similar sensitivity study for $CH_4$ resulted in errors of less than 0.02% for surface temperature changes on the order of 18 K. This is significantly smaller than the <1% diurnal variations in $CH_4$ in the SoCAB.

**Appendix C: Sampling Bias**

The sampling strategies of the MkIV and Caltech FTS measurements differ significantly. MkIV observations are performed manually. From JPL, the MkIV measures within one hour of solar noon, once or twice per week. The Caltech instrument is automated and measures throughout the day, every day, whenever it is sunny. To determine whether biases caused by these sampling differences affect the results of our analyses, we selected a subset of the coincident time series from MkIV and Caltech in 2015. We then filtered both datasets according to Appendix A, and further subselected the Caltech data to points within 15 minutes of the MkIV measurements. The grey dots in Figure C1 show all the filtered Caltech data; black diamonds are the Caltech data time-matched with MkIV; red circles are the MkIV data themselves. Slopes of the tracer-tracer anomalies in the third panel below show a small bias between the filtered and time-matched Caltech slopes, both well within the uncertainties of the MkIV slope. Thus, there should not be a significant bias introduced into the tracer-tracer slopes from the sampling strategy.

*Acknowledgements.* Part of this work was performed at the Jet Propulsion Laboratory, California Institute of Technology, under contract with NASA. We thank the various people who have assisted with MkIV ground-based observations over the years, and the NASA Upper Atmosphere Research Program for funding. This research was supported by NASA's Carbon Cycle Science program (NNX14AI60G). TCCON data were obtained from the TCCON Data Archive, hosted by the Carbon Dioxide Information Analysis Center (CDIAC) - tccon.onrl.gov.

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

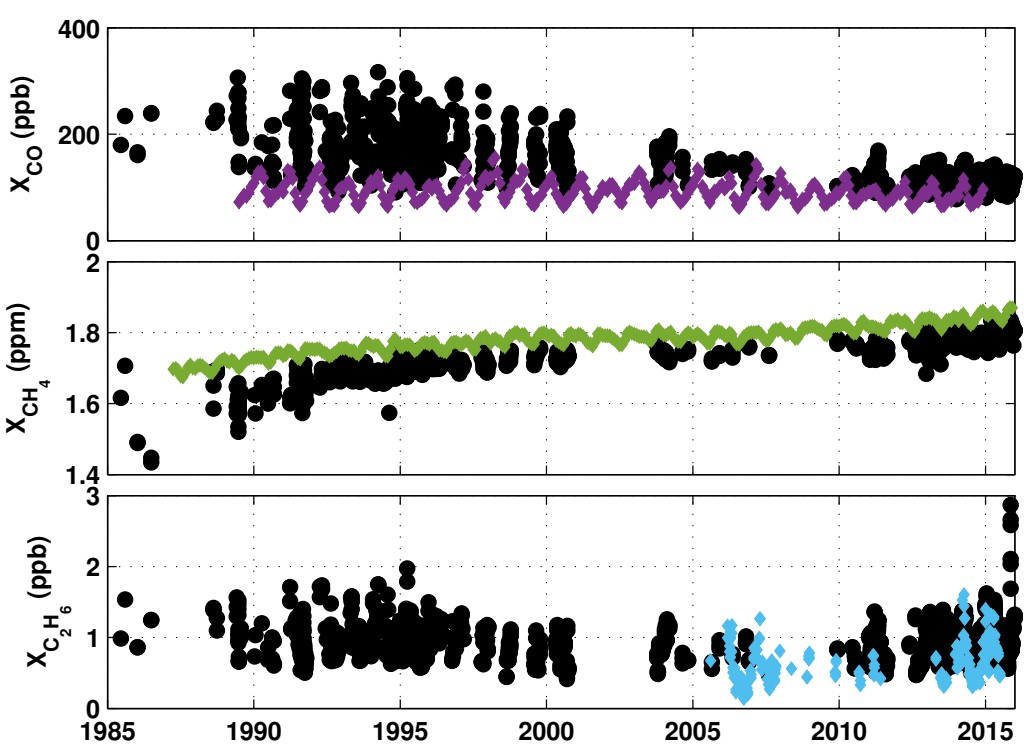

**Figure 1.** Time series from the MkIV FTS in the SoCAB. The colourful diamonds are the background surface *in situ* values measured atop Mauna Loa. The black circles indicate the MkIV FTS measurements of $X_{CO}$ (top), $X_{CH_4}$ (middle), and $X_{C_2H_6}$ (bottom). There is a marked decrease in both the day-to-day variability and median value in $X_{CO}$ over time, an increase in $X_{CH_4}$ in line with the global trends, and non-monotonic, seasonal changes in $X_{C_2H_6}$.

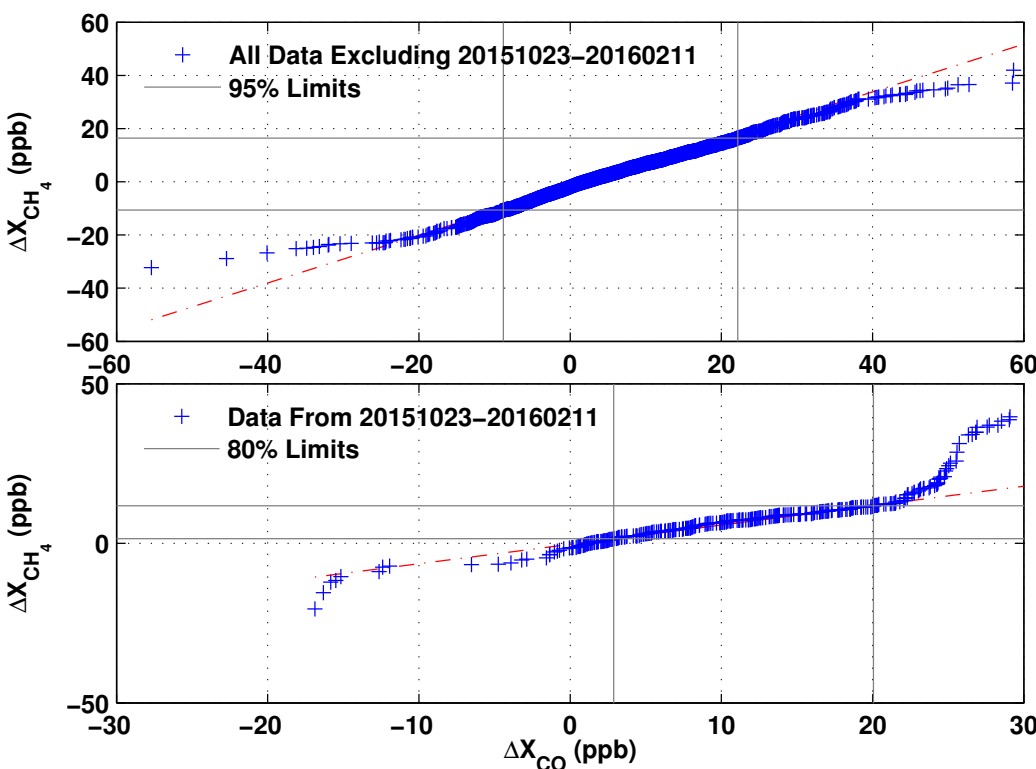

**Figure 2.** These quantile-quantile plots show the extent to which $\Delta X_{CH_4}$ and $\Delta X_{CO}$ anomaly data from the Caltech FTS are from the same probability distribution. When the distributions of the two datasets are similar, the points (blue '+') fall along the red dashed line. The top panel shows the quantile-quantile plot of methane and carbon monoxide from data prior to the Aliso Canyon gas leak, which started on October 23, 2015. The plot is linear between the grey lines which indicate the 95% quantiles of $\Delta X_{CO}$ and $\Delta X_{CH_4}$. We use these limits to define air that is representative of "ambient" SoCAB air from air that contains plumes during that time period. The bottom panel shows the quantiles of $\Delta X_{CH_4}$ and $\Delta X_{CO}$ anomaly data for the time period after the Aliso Canyon gas leak began. For this time period, 80% quantiles were chosen to distinguish between ambient and plume air.

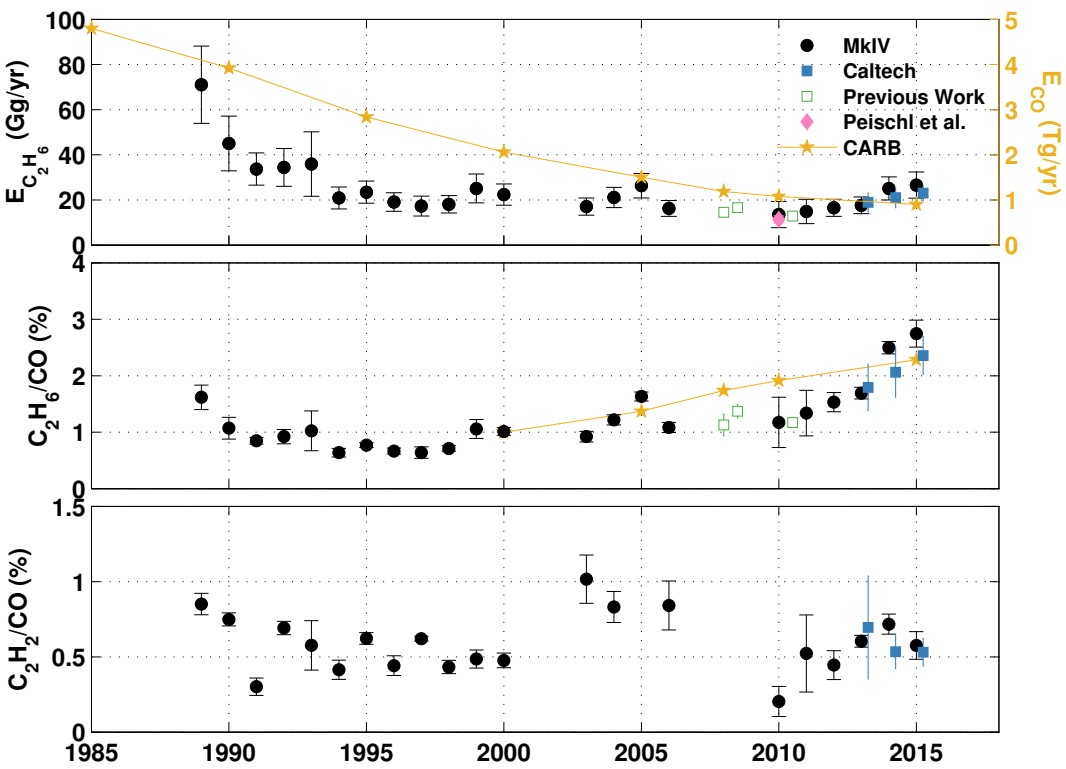

**Figure 3.** The top panel right axis shows the estimated carbon monoxide emissions inventory for the SoCAB, published by the California Air Resources Board (CARB). The top panel left axis shows the inferred emissions of ethane from the MkIV FTS (black circles), the Caltech FTS (blue squares), previous estimates from Wennberg et al. (2012) and Hsu et al. (2010) (green squares), and Peischl et al. (2013) (pink diamond). The second panel shows the ethane to carbon monoxide anomaly slopes from the MkIV FTS (black circles), the Caltech FTS (blue squares) and previous studies (green squares). The gold line with gold stars represents what the ethane to carbon monoxide anomaly slope would be if ethane in the atmosphere remained constant at 1% of the year 2000 carbon monoxide emissions from 2000 onward. The third panel shows the acetylene to carbon monoxide anomaly slopes, which are reasonably invariant over the time series.

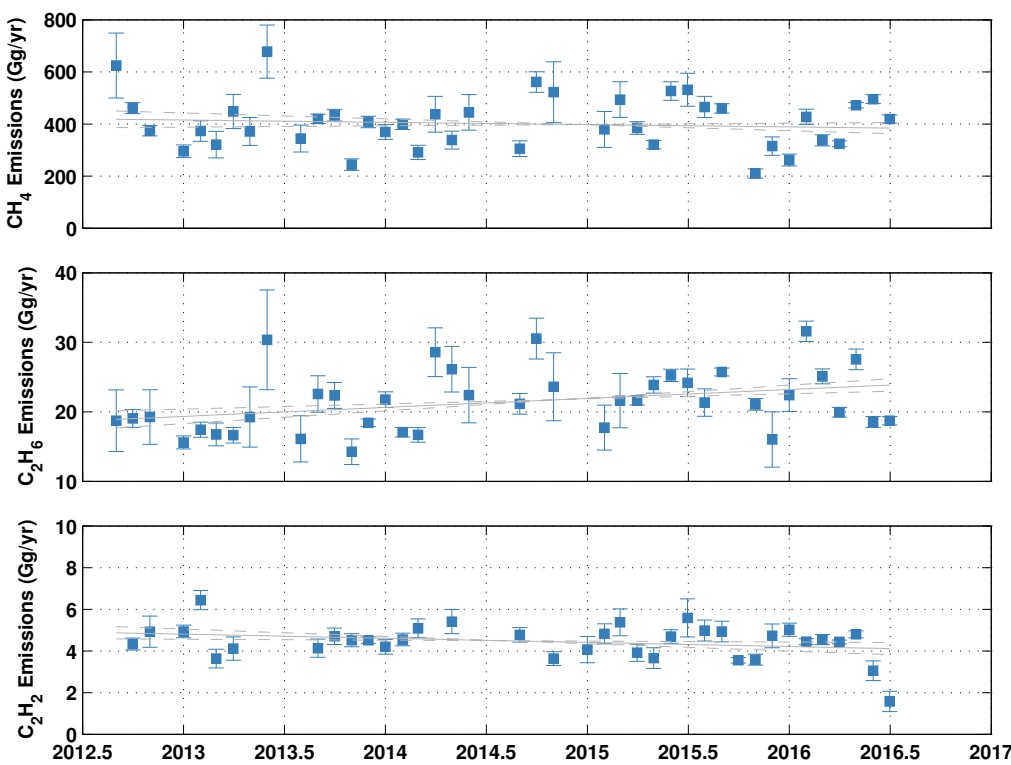

**Figure 4.** This plot shows the monthly methane (top), ethane (middle), and acetylene (bottom) emissions measured in the atmosphere by the Caltech FTS (blue squares). Grey solid lines indicate the best-fit slopes with standard errors indicated by the grey dashed lines.

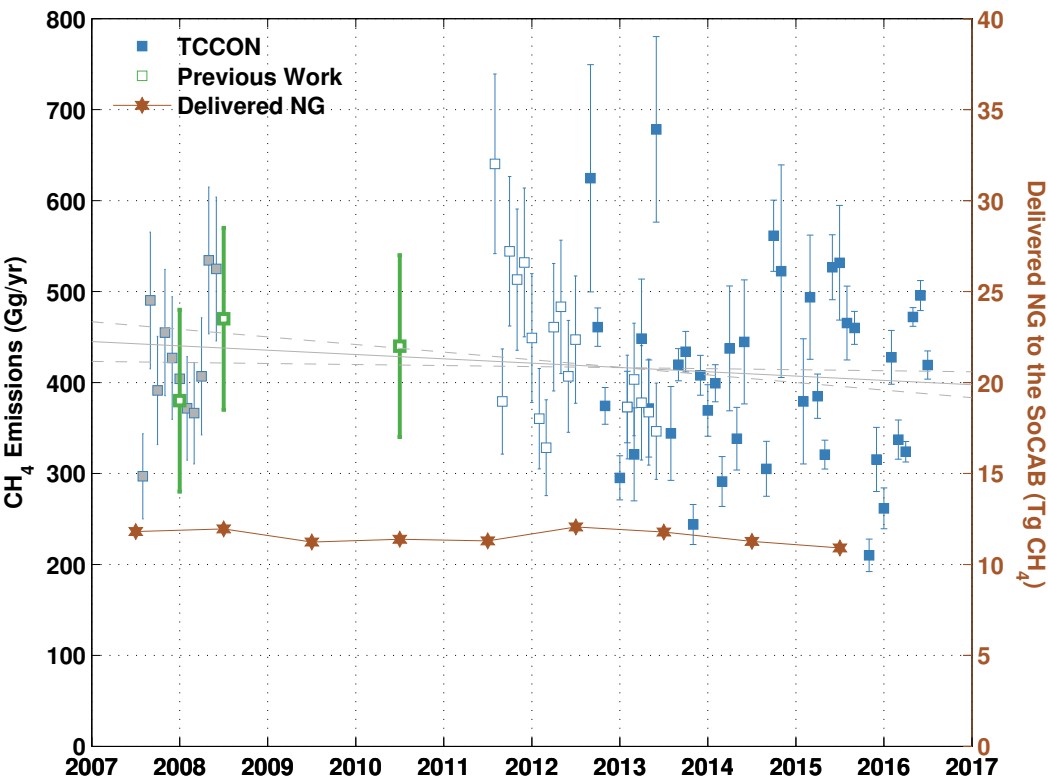

**Figure 5.** The left-hand axis shows methane emissions measured in the atmosphere by three TCCON FTS instruments that were located in the SoCAB since 2007. The grey solid line indicates the best-fit slope with standard errors indicated by the grey dashed lines. Previous measured emissions are indicated by green squares. The right-hand axis shows the delivered natural gas to the SoCAB, and is scaled such that if 2% of the delivered gas is released into the atmosphere, the atmospheric burden would be equal to the numbers (in Gg) on the left-hand axis.

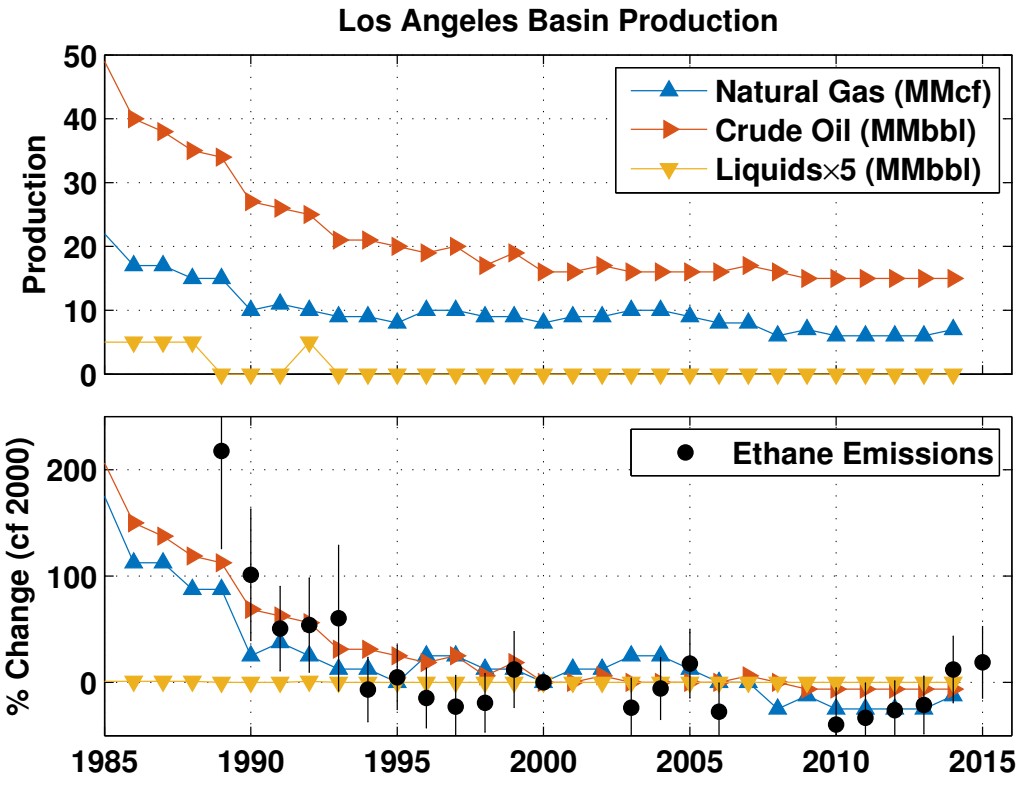

**Figure 6.** Natural gas, crude oil, and natural gas liquids production in the Los Angeles Basin, reported by USEIA (2015c, a, b) are shown in the top panel. The natural gas liquids production values are multiplied by 5 for scale. In the lower panel, the production is scaled to illustrate the changes in production relative to 2000. The MkIV $C_2H_6$ emissions relative to 2000 (black circles) are added for reference.

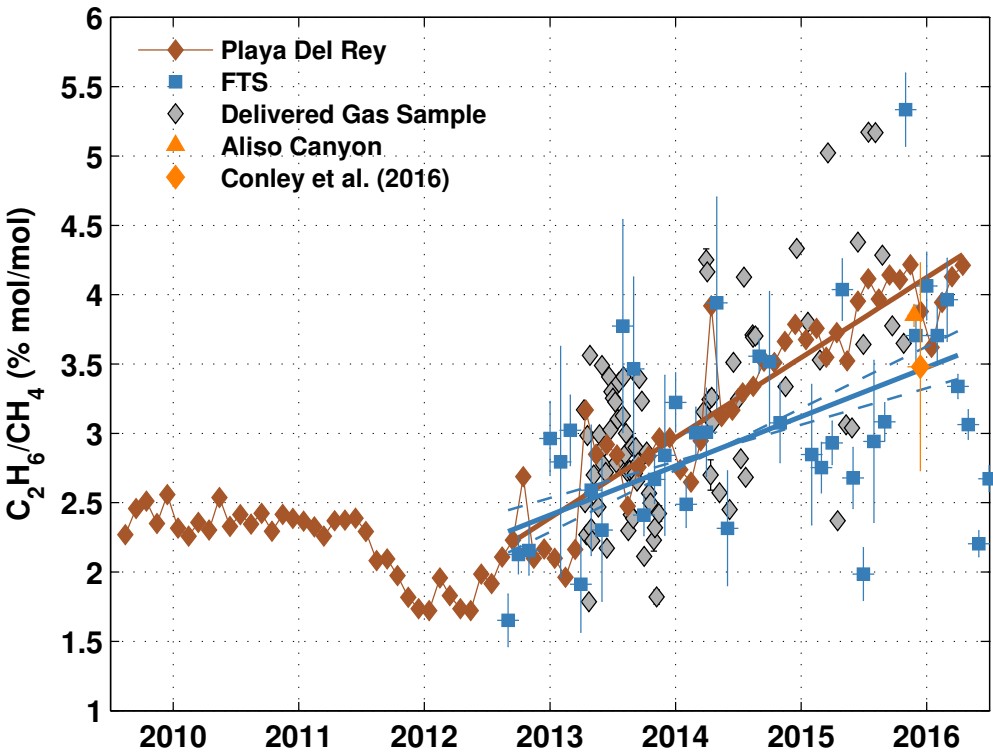

**Figure 7.** This time series shows the ethane to methane ratios in the Playa Del Rey gas storage facility (brown diamonds), in the natural gas delivered to the laboratory (grey diamonds) and in gas anomalies measured with the Caltech FTS (blue squares). The slope of the Playa Del Rey ratios is shown in brown; the slope of the Caltech FTS ratios is in blue with dashed lines indicating the slope uncertainty. The slope of the delivered gas samples is not shown, but is statistically indistinguishable from the Playa Del Rey slope. The median ethane to methane anomaly ratio measured by SoCalGas in the air near the Aliso Canyon gas leak is indicated by the orange triangle, and the value near Aliso Canyon measured from an aircraft platform by Conley et al. (2016) is indicated by the orange diamond.

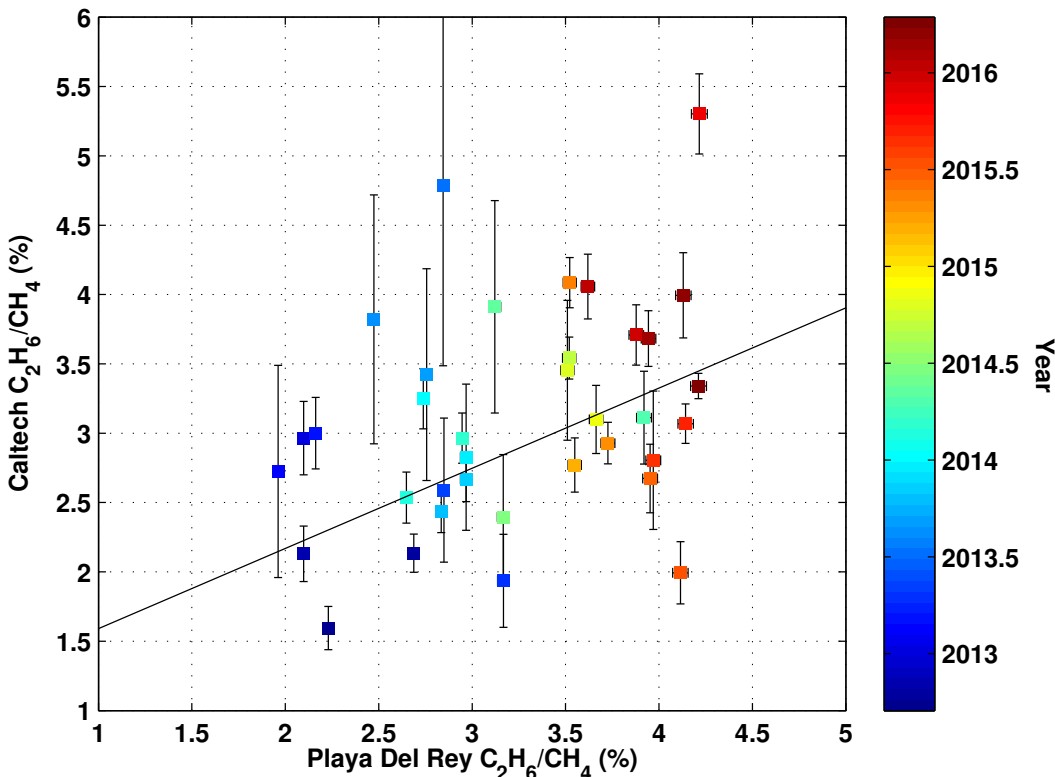

**Figure 8.** This figure shows the ethane to methane ratios from the Caltech FTS data on the y-axis and from the Playa Del Rey gas storage facility on the x-axis between September 2012 and March 2016. The colours indicate the date of the measurements. The slope of the relationship is indicated by the black line $(0.58 \pm 0.12)$ and is consistent with the slope derived from Figure 7.

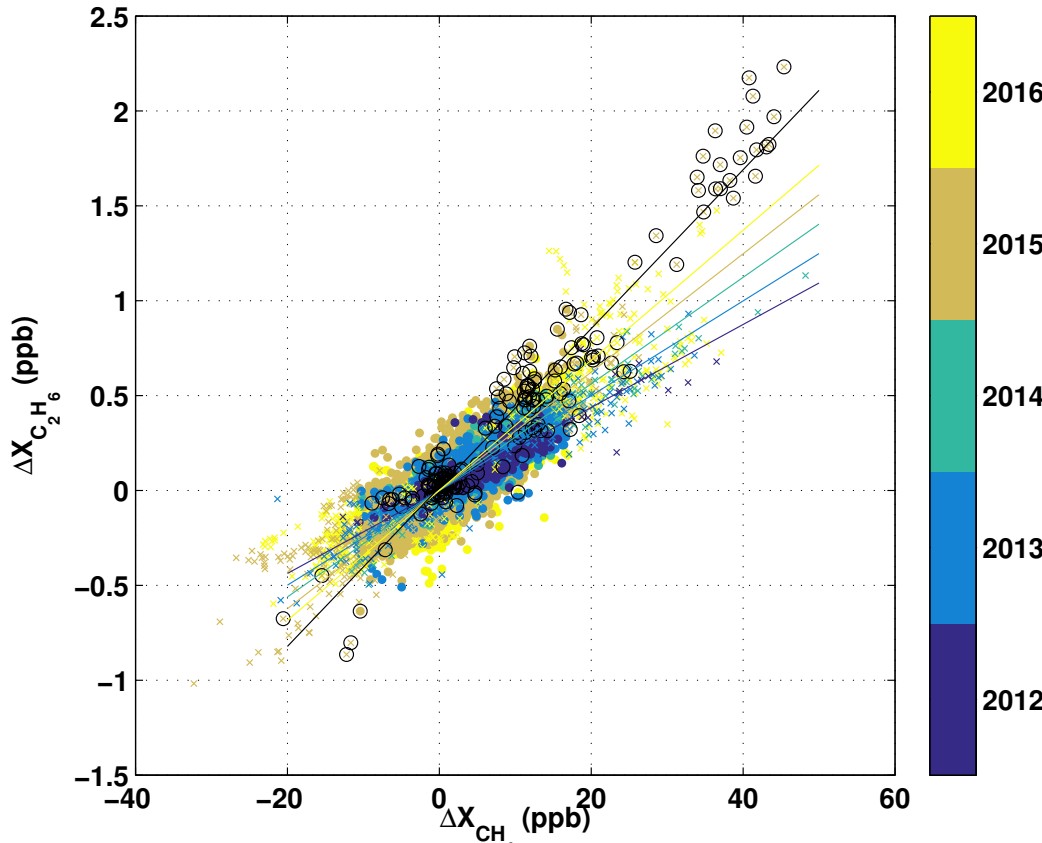

**Figure 9.** This figure shows the ethane and methane anomalies during the Caltech TCCON record. The entire time series is represented by filled circles, the plume data is represented by 'x' symbols, and the measurements of the plume originating from the Aliso Canyon gas leak are circled in black. The colours represent the year during which the measurements were recorded. The average ambient slopes from Figure 7 are indicated with solid lines, and show a time dependence consistent with the slopes from plumes. The ethane to methane slope in the Aliso Canyon plume data (black line) shows a high degree of correlation ($R^2 = 0.95$) and a slope of $4.28 \pm 0.07\%$. Note that the ethane to methane ratios in the ambient air were rising throughout the record.

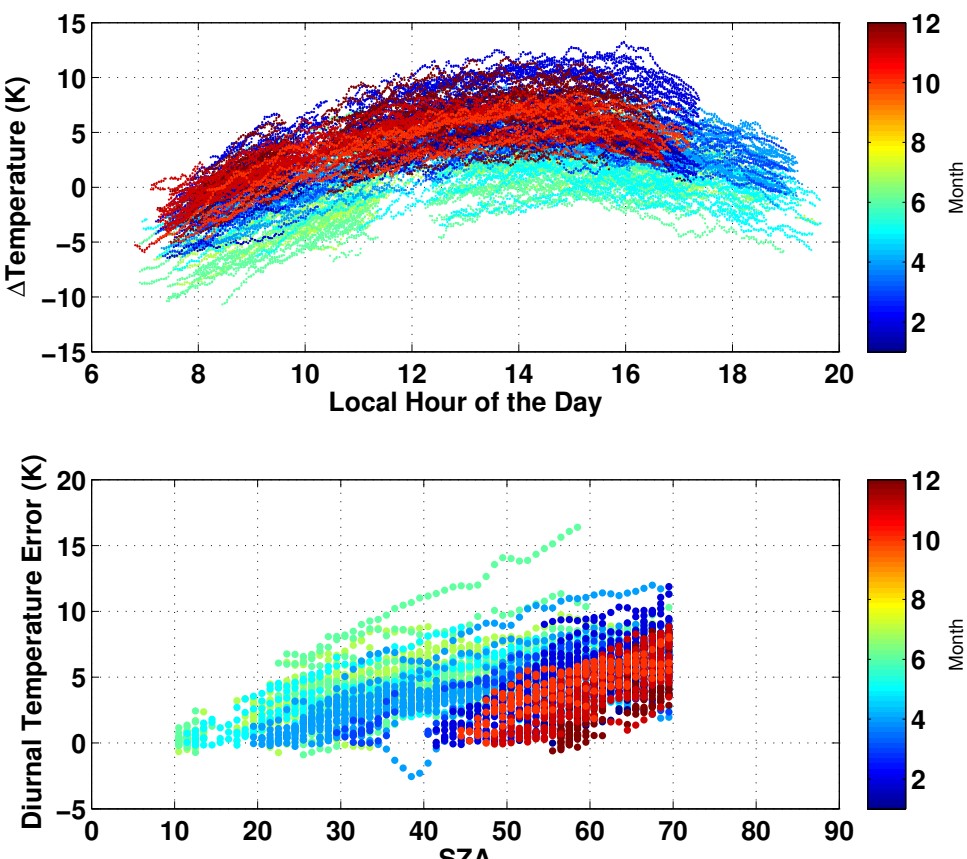

**Figure B1.** This figure shows the change in surface temperature throughout the day from the noontime a priori value (top panel), and the diurnal surface temperature error in the bottom panel. The diurnal surface temperature error is computed by subtracting morning from afternoon surface temperatures in the same way as the trace gas anomalies are computed.

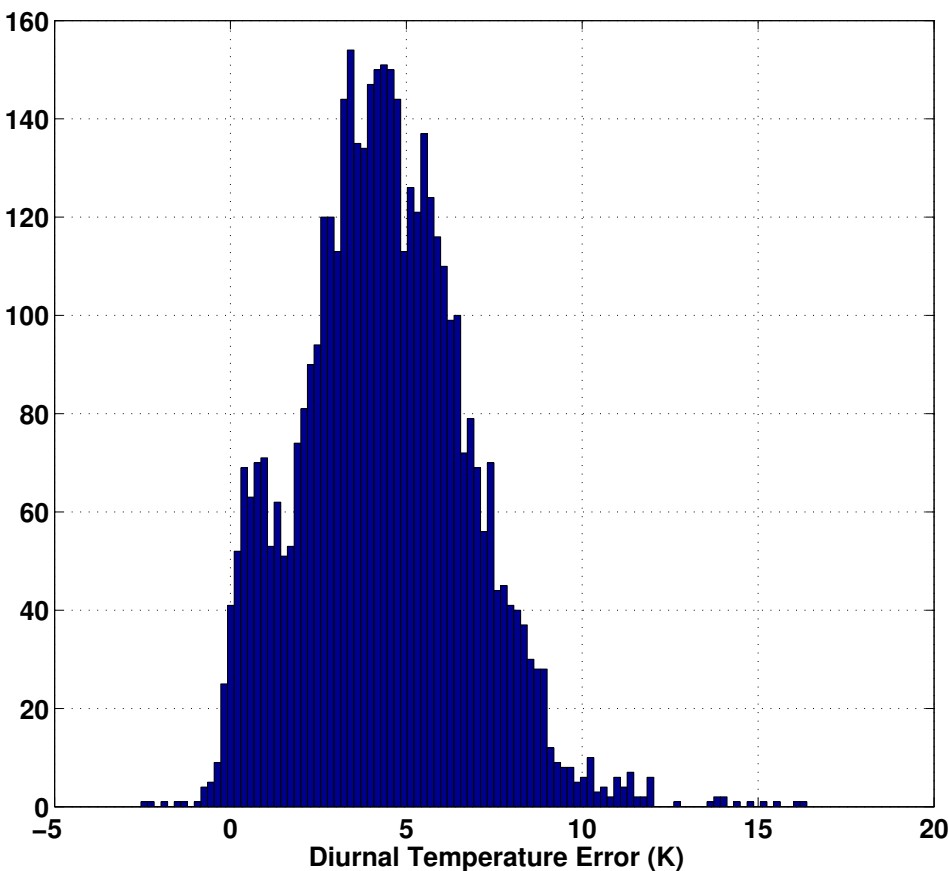

**Figure B2.** This figure shows a histogram of the surface temperature anomalies shown in the bottom panel of Fig. B1.

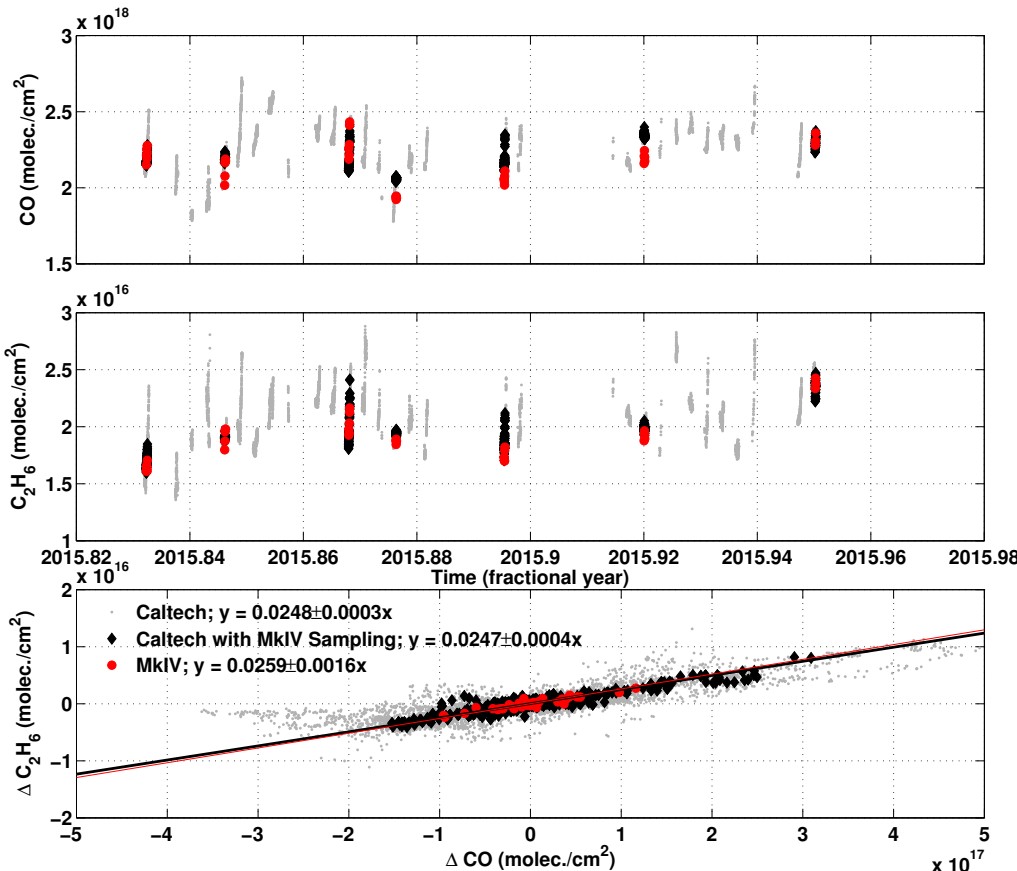

**Figure C1.** This figure shows the negligible impact on the derived tracer-tracer slope from the different sampling strategies used by the MkIV and Caltech measurements. The top panel shows the time series for CO total column abundances for Caltech (grey), MkIV (red), and the subsampled Caltech values to coincide with the MkIV measurements (black). The middle panel shows the $C_2H_6$ time series. The bottom panel shows the tracer-tracer relationship between the diurnal anomalies of the trace gases. The slopes computed for the three cases agree well within error.

**Table 1.** Emissions Inventories for $CH_4$ and $C_2H_6$. Only one methane emission value is included which is the mean emissions over the 2007-2015 measurement period. Ethane emissions are from the Caltech measurements only, and each column of the table contains data from September through August. Emissions marked with a dagger (†) are from Peischl et al. (2013) for 2010. The Pipeline Natural Gas emissions of ethane are computed by multiplying the methane emissions from the pipeline natural gas (58% of the measured total $CH_4$) by the increasing slope fitted to the ethane to methane ratios measured by the Caltech instrument. Uncertainties on the "Measured" emissions are the standard deviations of the monthly emissions computed for the time range.

| Source | $CH_4$ Emissions Gg $CH_4 \cdot yr^{-1}$ 2007–2015 | $C_2H_6$ Emissions Gg $C_2H_6 \cdot yr^{-1}$ 2012-2013 | 2013-2014 | 2014-2015 |
|---|---|---|---|---|
| Biogenics | $182\pm54^{†}$ | — | — | — |
| Local Oil and Gas | $32\pm7^{†}$ | $4.5\pm1.0^{†}$ | $4.5\pm1.0^{†}$ | $4.5\pm1.0^{†}$ |
| Vehicles and "Other" | — | $0.9\pm0.1^{†}$ | $0.9\pm0.1^{†}$ | $0.9\pm0.1^{†}$ |
| Pipeline Natural Gas | $240\pm73$ | $11.6\pm4.4$ | $13.3\pm5.0$ | $15.0\pm5.7$ |
| Inventory Total | $453\pm91$ | $17.0\pm4.5$ | $18.7\pm5.1$ | $20.4\pm5.7$ |
| Measured | $413\pm86$ | $19\pm4$ | $21\pm4$ | $23\pm3$ |