# Peer review of "Quantifying the Loss of Processed Natural Gas Within California's South Coast Air Basin Using Long-term Measurements of Ethane and Methane"

_Atmospheric Chemistry and Physics, 2016_

## Referee Comment (RC1) · Anonymous Referee #1 · 14 Jun 2016

The work under consideration "Quantifying the Loss of Processed Natural Gas..." by D. Wunch et al. investigates the origin of increased methane emissions in California's SoCAB (South Coast Air Basin) region using remote sensing observations of carbon monoxide, ethane, and methane. Overall, the paper is very well written and fits nicely within the scope of ACP. I recommend publication, but would like to ask the authors to address a couple of issues in a revised version of the manuscript.

Observations from two different spectrometers are used in this work, the MkIV spectrometer covering a much longer period, but with sparser observations, and the Caltech TCCON spectrometer. I am a bit concerned with respect to the compatibility of the derived gas anomalies. The point is not primarily the spectral measurement itself, but the

significantly different sampling strategies of the measurements (MkIV observations are constrained to local noon). It would be instructive to demonstrate that an analysis of a reduced TCCON dataset (local noon observations only) generates compatible gas anomaly values, or whether the sampling strategy can introduce a significant bias.

The determination of gas anomaly values from the difference of afternoon and morning values is in principle a convincing approach. However, as small changes are derived from differences of much larger column values, I wonder whether the heating of the boundary layer during the day might also mimic a gas anomaly contribution? Is the analysis performed assuming a constant temperature profile? Is the heating effect a significant disturbance?

I have problems to understand that in Figure 2 the slope of the red dashed line differs between the top and bottom panels. If the slope is a function of time, why then is the slope in the upper panel so well defined (it encompasses data from several years, correct?).

The error bars on the symbols in Figure 3 are difficult to read. It seems that while the 2013 and 2015 results from MkIV and the TCCON spectrometer agree nicely, the discrepancy in 2014 is much larger than the indicated error bars. Is this a sampling issue (dates of observations used?)

The scatter of the FTS deduced ethane to methane ratios in Figure 5 is large. The error bars on the individual data points are quite variable and especially in 2015, the scatter between the data points is much larger than the individual error bars. Why? Does this imply that the uncertainty budget is dominated by a sampling statistics issue? What is the level of significance for the derived slope value? Does the regression fit take into account a weighting of data points accoring to the individual error bars? The figure might suggest a superimposed peak of high ratio values in the mid of 2013.

In Figure 7, the claimed steady rise of the slope during the observation period is hardly recognizable (due to the overlap of data points), perhaps a subdivision in several panels spanning fractions of the whole period would improve the readability.

---

## Short Comment (SC1) · 14 Jun 2016

The observed increase in methane and ethane emissions may be partially attributable to decreased oxidation of methane and ethane by soil microbes. The surface flux of natural gas leaks can be reduced by microbial oxidation. GRI/EPA 1996 reports up to 40% of leak emissions can be oxidized within the soil. Several factors including moisture content and temperature affect the methane oxidation rate of the soil microbial community. Van den Pol-van Dasselaar 1998 report that methane oxidation in sandy grassland soils is highest at intermediate soil moisture and ceases below 5% moisture content. The severe, extended drought in southern California since 2012 might cause local distribution emissions to increase if inhibited microbial oxidation allows a greater

fraction of underground leak emissions to reach the surface. It is possible that decreased microbial oxidation may also increase emissions from geologic seepage and biogenic sources. I recommend that you address this issue in your discussion.

http://link.springer.com/article/10.1007/BF00425043 https://www.epa.gov/gasstar/documents/emissions_report/9_undergr
http://link.springer.com/article/10.1023/A:1004371309361
http://droughtmonitor.unl.edu

---

## Referee Comment (RC2) · Anonymous Referee #2 · 15 Jun 2016

This manuscript presents two independent time series of column measurements of methane (C1) and ethane (C2) in Southern California and uses them to infer trends in annual emissions in the region. The manuscript is generally well-written and I would recommend publication with targeted additions and clarifications as further described below. In particular, the manuscript would be improved by providing a more systematic discussion of the causes of recent changes in C1 and C2 emissions.

General comments

The authors should consider and discuss the statistical significance of the reported trends in observed C1 and C2. The confidence intervals around the annual averages in Table 1, for example, suggest the annual averages across the 3-years shown are

not statistically different. On the other hand, assuming the error bars shown in Figure 3 are correct, the 2015 values for C2 emissions seem to be statistically higher than those during 2006-2010. The authors should consider whether the monthly C1 and C2 emission time series in Figure 6 provide an alternative basis to determine the existence of a significant trend (e.g., are the slopes statistically different than zero?).

Wunch et al (2009) used CO2 instead of CO as the basis to estimate CH4. Also, Wunch et al (2009) pointed out the possible underestimation of CH4 if it was computed from CO emissions, given their differing diurnal profiles (CO emissions primarily influenced by traffic, which was believed to be a stronger daytime source than methane). This new discussion manuscript does not address these issues. The authors should clarify how potential differences in the diurnal profiles of CO, CH4, and C2H6 could affect the emissions estimates calculated with Equation 4.

The authors should emphasize the importance to their analysis of the changing C2:C1 ratio in pipeline gas. This trend appears to serve as tracer of opportunity, a unique fingerprint that allows attribution of the total observed C1 signal to infrastructure associated with handling, storage, delivery and use of pipeline quality natural gas. This is done indirectly in line 220, but the scientific novelty and utility of the trend deserves greater attention.

The manuscript's impact would be improved if the authors could provide a more complete picture about the contribution of specific source types to the observed C1 and C2 trends. Having partitioned the fraction of total methane signal due to pipeline gas (possible due to its increasing ethane content), can the authors further delve into the individual methane and ethane trends and provide a conceptual model that explains the recent trends or patterns in monthly/annual C1 and C2 emissions. [It would seem the C2 emissions might be reducible to a 2-source model (pipeline gas and associated gas/geologic seepage) with appropriate adjustment for vehicle emissions. Similarly, C1 emissions might be reducible to a 3-source model, by adding a generic third term for biogenic C1 sources.] At a minimum, the authors should clearly indicate whether the

increasing C2:C1 ratio in pipeline gas is, by itself, sufficient to explain the potentially increasing C2 trend in Fig 3 and 6? Or can the balance of the C2 budget not explained by pipeline gas losses be explained: for example, given likely associated gas compositions, could the local oil/gas production to which Peischl attributed 32 Gg C1 also account for the excess C2 that is not explained by losses of pipeline quality gas? Alternatively, are other causes required?

Once the C2 budget is determined, and knowing the C2:C1 ratio of pipeline gas, what can the authors say about the trend in C1 emissions due to losses of pipeline quality gas? It would be valuable if the authors could provide an assessment of whether the data indicates that downstream natural gas emissions in the region are changing.

It is not clear from the text at line 225 and the reference cited how the authors derive the mass of C1 delivered by SoCalGas to customers within the SoCAB. It is also unclear why sales data going back to 2003 are relevant at this point of the discussion focused on the regional methane budget in 2015 (it would be more relevant – indeed desirable – to show historical gas deliveries in Figure 4). Southern California Gas' annual report (Sempra Energy 2015 Financial Report) reported annual volumes of gas sold in 2013, 2014 and 2015 of 999, 944 and 925 bcf (average 960 bcf, or 17.5 Tg assuming a methane content in gas of 95% ). The authors should explain how they partition the SoCalGas' systemwide sales to isolate the customers solely within the SoCAB. Because not all of the gas sold by SoCalGas is consumed within the SoCAB and may or may not be transported through the SoCAB, the authors should report multiple metrics for the loss of pipeline quality gas that is sold or transported across the basin. One metric would be % of methane delivered that is emitted, and the other is the emissions as a percent of methane throughput in the SoCalGas system. The latter yields a loss rate for pipeline gas of 1.4% of potential throughput (242Gg/17.5Tg). The comparison to Wennberg et al's 2% loss rate should be done with caution, ensuring that the quantities in the numerator and denominator are apples-to-apples between this work and the previous work (it seems the 2% in Wennberg would most appropriately be compared

to 1.4%, as calculated above).

Figure 4. The manuscript would be improved if the hydrocarbon production data provided was specific to the SoCAB rather than statewide (these are publicly available from state agencies). Additionally, since hydrocarbon production is only a small contributor to C1 and C2 emissions in the SoCAB, this figure would be much more useful if it presented publicly available activity trends for other chief sources – in particular, I would suggest SoCalGas' natural gas sales and livestock populations. Recent CH4 emissions data or landfills and waste water treatment plants may also be available through the US EPA Greenhouse Gas Reporting program or California state equivalents.

The richest findings seem to derive from the more recent and denser Caltech FTS measurements, with the JPL MkIV FTS data providing corroboration and further insight about historical trends. The manuscript's flow and clarity might be improved with some reorganization of the results and discussion or more explicit delineation of how the two data sets are used to support the conclusions reached.

The results relating to Aliso Canyon are interesting and important, but are not central to the paper's main findings. I would recommend moving the Aliso Canyon discussion into a separate subsection.

Detailed comments

Abstract Line 9. The introduction of "Our methane emissions record" here is confusing since line 4 refers to a record dating back to the 1980s.

Abstract Lines 10-15. This wording might be misconstrued to imply that the source of the excess methane is the gas storage facility. In fact the gas storage facility is only mentioned since it is a reliable source of C2:C1 ratios. But the authors have a secondary data source (delivered gas) that yields a statistically indistinguishable trend line in Fig. 5. The authors should revise the language to indicate the comparison is between atmospheric measurements and measured C2:C1 of gas delivered and stored in

the region. Additionally, the authors should more explicitly indicate the scope of natural gas infrastructure implicated in the final sentence – to indicate it includes gas delivery infrastructure including pipeline leaks (transmission and distribution), compression and storage facilities, and post-meter losses among others.

Line 179. It was unclear how the statement about ethane to acetylene ratios followed from statements about C2:CO and acetylene:CO; please elaborate on the significance.

Line 226. The statement attributing 242 Gg/yr C1 to natural gas infrastructure should be linked back to the prior paragraph's finding that 54% of total excess was due to natural gas (e.g. "242 Gg/yr, equal to 54% of the SoCab total. . .".

Lines 248-255. The specific value used for GWP100 should be stated (e.g., 25, 28, or 34). The choice of 100-yr GWP in this paragraph does not account for the greater short-term climate impacts of CH4. The authors should consider reporting a 20-yr CO2e value in addition to the 100-yr value. The reference to climate impact in the last sentence needs to explicitly distinguish short- and long-term impacts; if only 100-yr GWP comparisons are made, then the sentence should be clarified to refer to "long-term climate impact. . ."

Figure 1. The very rapid rise in C2 mole fraction in the most recent JPL MkIV FTS measurements should be explained (panel 3). Is this trend due to the increased C2:C1 ratio, the Aliso Canyon blowout or both? Should the C2 rise be accompanied by changes in C1?

Editorial Comments

Line 141. The word "are" appears twice.

Figure 1. The black Mauna Loa data points are significantly obscured by the CO and C2 data points.

Fig 3. The error bars are hard to make out and the symbol for the Peischl et al is not evident.

[Figure]

---

## Referee Comment (RC3) · Anonymous Referee #3 · 17 Jun 2016

The manuscript presents an interesting new result and is generally well-written. I would recommend publication with some revisions to clarify several points.

Comments on the text In the paragraph starting on line 187, the authors relate the extraction of petroleum from the SoCAB to the production in the rest of the state. This seems likely to be a valid assumption, but it would be helpful here to provide some additional justification. Would the results of the analysis be substantially different if it is assumed that SoCAB petroleum extraction tracked regional or national trends? Lines 230-235 discuss how non-petroleum sources can close the methane budget. It would be helpful to discuss changes in these sources here to corroborate the conclusion that petroleum accounts for only half of the observed methane increase.

Comments on the figures The panels on Figures 2 and 3 have "squashed" aspect ratios that make them slightly difficult to read. The bottom panel of Figure 2, for example, compresses much of the data into a small region of the graph. In Figure 3, the presence of four panels in a single figure makes it difficult to see the trends described in the caption. Could some of these panels be merged and their axes modified to make the graphs taller? The error bars on the atmospheric ratios in Figure 5 are quite large and imply a large uncertainty in the calculated slope. Indeed, this uncertainty is reflected in the text as well. A visualization of this uncertainty in the figure would be beneficial. Line 219 reports the ratio of slopes as $54 \pm 20\%$, which is thereafter referred to as "about half." However, the large uncertainty in the slope means that the atmospheric increase could be anywhere from not well explained by the changing storage ratios (about 1/3), to very well explained (over 2/3). Do the authors have speculation as to whether the percentage is on the high or low end of this range?

Editorial comments In line 244, the slope of the ethane/methane correlation is $4.28 \pm 0.07\%$. This piece of information is in agreement with the storage "ratios exceeding 4%" in line 209. I suggest placing these pieces of information closer together to emphasize this connection, because it provides further evidence that the Aliso Canyon plume was detected. The uncertainties are reported in an inconsistent manner in the text. Line 8 of the abstract contains the quantities $13 \pm 4.5$ and $25.8 \pm 3.9$; and line 234 of the text contains the quantity $32 \pm 7$. Some further discussion of how these different levels of uncertainty for these and other quantities reported in the text were chosen would be helpful.

---

## Referee Comment (RC4) · Anonymous Referee #4 · 21 Jun 2016

General comments

Wunch et al. derive emissions of ethane in the South Coast Air Basin dating back to the late 1980s. They further derive emissions of methane from 2012 to late 2015. They use two Fourier Transform Spectroscopy instruments to make measurements of atmospheric enhancements of ethane, methane, carbon monoxide, and acetylene, along with a South Coast Air Basin inventory estimate of carbon monoxide emissions, to derive emissions of ethane and methane. They also use ethane/methane enhancement ratios and their abundance in natural gas delivered to the region to determine that approximately half the methane emissions are due to leaking natural gas infrastructure.

Overall, this paper is well-written and the analysis is interesting, pertinent to megacity

greenhouse gas emissions, and the conclusions are mostly well-supported. However, some of the conclusions seem to be at odds with another paper currently submitted to ACPD, Wong et al., that concludes that methane emissions in the SoCAB have been decreasing since 2011, albeit with a low confidence interval. Some discussion comparing and contrasting the conclusions of Wong et al. is warranted. For instance, how well does the Caltech FTS represent the entire SoCAB methane emission, compared to the multiple measurement locations described by Wong et al.?

Specific comments

Line 42, the sampling location of Hopkins et al. and Townsend-Small et al. were heavily skewed toward the western SoCAB. How well do those studies represent emissions to the entire region?

Line 110, why do you subtract the daily mean of ethane, CO, and acetylene and not the lowest value?

Line 111-113, by aggregating for an entire year, how do you account for this slope not representing the seasonal variability instead of variability due to emissions?

Line 185, is there an earlier reference you could use to support your conclusion that ethane emissions from automobiles would not have accounted for the emissions decline in the late 1980s? The conclusions from the mid-90s on are well supported, but it is unclear they are relevant to the 1980s.

Line 236, can you confirm with your data that the Aliso Canyon leak did not occur before October 23? There have been some reports of skeptical homeowners questioning that it may have been leaking before this date.

Line 245, is the ethane emission from Aliso Canyon found by multiplying the 4.28% anomaly by the Conley et al. methane emission of 97.1 Gg? If so, this should be stated more clearly.

Line 250, please state which 100-yr global warming potential you used. 25?

Line 262, what is the uncertainty of the ~20%? This would help in the comparison with Wong et al.

Technical comments

Line 72, equation 2, a subscripted "dry air" might fit better for the "column dry air", similar to how it is done for the molecular mass?

Line 143, Conley et al. state the facility has a capacity of 168 billion cubic feet, and a "working capacity" of 86 billion

Line 151-152, Suggest swapping "near the facility" and "from aircraft"

Line 167, change "represents" to "represent"

Line 282, please define "HF"

Figure 4 might look "cleaner" if you used the daily average production for a given month. The variability of the days in a month results in a ~3% noise, which is close to the noise between 2003 and 2010.

---

## Author Comment (AC1) · 9 Sep 2016

Response to David Lyon.

Referee comments are in *red italics*, our responses are in black text.

*The observed increase in methane and ethane emissions may be partially attributable to decreased oxidation of methane and ethane by soil microbes. The surface flux of natural gas leaks can be reduced by microbial oxidation. GRI/EPA 1996 reports up to 40% of leak emissions can be oxidized within the soil. Several factors including moisture content and temperature affect the methane oxidation rate of the soil microbial community. Van den Pol-van Dasselaar 1998 report that methane oxidation in sandy grassland soils is highest at intermediate soil moisture and ceases below 5% moisture content. The severe, extended drought in southern California since 2012 might cause local distribution emissions to increase if inhibited microbial oxidation allows a greater fraction of underground leak emissions to reach the surface. It is possible that decreased microbial oxidation may also increase emissions from geologic seepage and biogenic sources. I recommend that you address this issue in your discussion.*

*http://link.springer.com/article/10.1007/BF00425043*
*https://www.epa.gov/gasstar/documents/emissions_report/9_underground.pdf*
*http://link.springer.com/article/10.1023/A:1004371309361*
*http://droughtmonitor.unl.edu*

We thank Dr. Lyon for bringing this issue to our attention. It seems likely to be a small effect. A brief discussion has been added to the paper:

Droughts such as the one plaguing Southern California since 2012/2013 [Swain2014,Griffin2014] can reduce the ability of soil microbes to remove methane and ethane released underground into the soils [vandenPol-vanDasselaar1998,Adamse1972]. The constant $CH_4$ emissions and growing $C_2H_6$ emissions since 2012 would require a compensating decrease in biogenic emissions of $CH_4$ to offset this effect. However, biogenics are reported to have decreased by about 1% between 2012 and 2014 [CARB], so this effect is likely to be small.

---

## Author Comment (AC2) · 9 Sep 2016

Response to Anonymous Referee #1

We thank the referee for their valuable comments, which substantially improved the paper.

Referee comments are in *red italics*, our responses are in black text.

*Observations from two different spectrometers are used in this work, the MkIV spectrometer covering a much longer period, but with sparser observations, and the Caltech TCCON spectrometer. I am a bit concerned with respect to the compatibility of the derived gas anomalies. The point is not primarily the spectral measurement itself, but the significantly different sampling strategies of the measurements (MkIV observations are constrained to local noon). It would be instructive to demonstrate that an analysis of a reduced TCCON dataset (local noon observations only) generates compatible gas anomaly values, or whether the sampling strategy can introduce a significant bias.*

The figure below demonstrates that there should not be a significant bias induced into the tracer-tracer slopes. Shown below are a subset of the time series from MkIV and Caltech, filtered both datasets appropriately (removed plumes, cloudy data, etc.), and then subselected the Caltech data to points within 15 minutes of the MkIV measurements. The black dots below are all the filtered Caltech data; blue are the Caltech data time-matched with MkIV; red dots are the MkIV data themselves. Slopes of the tracer-tracer anomalies in the third panel below show a small bias between the Caltech and MkIV data that is well within the uncertainties in the slopes.

[Figure]

*The determination of gas anomaly values from the difference of afternoon and morning values is in principle a convincing approach. However, as small changes are derived from differences of much larger column values, I wonder whether the heating of the boundary layer during the day might also mimic a gas anomaly contribution? Is the analysis performed assuming a constant temperature profile? Is the heating effect a significant disturbance?*

The analysis uses a single *a priori* temperature profile throughout each day, that is representative of the local noon temperature profile, derived from the NCEP/NCAR reanalysis data. There is a systematic increase in surface temperature throughout the day; typically a 5K error between mid-morning and mid-afternoon at the surface (see histogram below); temperature changes aloft should be smaller and thus the integrated temperature error throughout the PBL should be smaller than 5K.

To minimize the temperature sensitivity of our retrievals, we chose windows in which the target absorption lines have average ground-state energies of around 300 cm-1. For example, we use the entire CO and CH4 bands in the near infrared, which have roughly the same number of high-j and low-j lines, reducing the temperature sensitivity. C2H6 is measured in its Q-branches between 2976 and 2997 cm-1. Based on performing C2H6 retrievals using correct and incorrect (perturbed) temperature profiles under a range of different conditions (temperature, humidity), we know that the retrieved C2H6 amount will change by <1% for a temperature perturbation of 5K at the surface, decreasing to zero at 3.5 km altitude. Since a typical diurnal change between mid-afternoon and mid-morning in the retrieved C2H6 is ~20%, the temperature-induced affect is comparatively small.

A sensitivity study for CH4 was performed by Hedelius et al. [2016] that showed <0.05% errors arising from 10K temperature perturbations between the surface and 700 hPa for lower resolution FTS instruments, and thus the sensitivity should be smaller for the higher-resolution TCCON instrument. This is also smaller than the <1% diurnal variations in CH4.

[Figure]

*I have problems to understand that in Figure 2 the slope of the red dashed line differs between the top and bottom panels. If the slope is a function of time, why then is the slope in the upper panel so well defined (it encompasses data from several years, correct?).*

The slopes of the red dashed lines in the quantile-quantile plots are not particularly important and the slope is not (necessarily) related to time. These plots were meant to show how we distinguished ambient SoCAB air from plumes. The bottom plot is for the time period when the Aliso Canyon leak was ongoing; the top panel is from the other time period. In this type of plot, data that are derived from a statistically similar set appear linear; that is, when the CO and CH4 vary simultaneously, their quantile-quantile plot will be linear. When they do not co-vary (i.e. when there is a plume of CH4),

their quantile-quantile plot will be nonlinear. We plotted the two times separately simply because the Aliso Canyon plumes will dominate the later data, and thus we wanted to be able to choose the filters that delineate between plumes and ambient air differently.

*The error bars on the symbols in Figure 3 are difficult to read. It seems that while the 2013 and 2015 results from MkIV and the TCCON spectrometer agree nicely, the discrepancy in 2014 is much larger than the indicated error bars. Is this a sampling issue (dates of observations used?)*

The Caltech annual mean slopes are calculated for September through August. If we calculate the mean 2014 monthly slope from the Caltech data for January-December, the slope increases from 2.1+/-0.4% (for September 2013 -August 2014) to 2.3+/-0.5%. (Closer to the 2.5+/-0.1% from MkIV.) We have also updated this figure to show the Caltech mean monthly slopes with the standard deviations as the uncertainties. (Previously we computed annual slopes and reported the slope errors, but we feel that was more complicated than necessary, and that the standard deviation of the monthly slopes provides a better estimate of the uncertainties.)

*The scatter of the FTS deduced ethane to methane ratios in Figure 5 is large. The error bars on the individual data points are quite variable and especially in 2015, the scatter between the data points is much larger than the individual error bars. Why? Does this imply that the uncertainty budget is dominated by a sampling statistics issue?*

Indeed, the variability in the FTS-deduced ethane to methane ratios is large. We would also point out that this is also true of the ratios in the delivered natural gas, which are very precise and accurate, and have very small error bars. This suggests to us that the delivered gas itself is quite a bit more variable than the reported withdrawn gas ratios from the Playa Del Rey storage facility, and our atmospheric measurements are able to detect that.

*What is the level of significance for the derived slope value? Does the regression fit take into account a weighting of data points accoring to the individual error bars?*

The slope of the ethane to methane ratios has an uncertainty of ~15%. The regression fit to the FTS data does take x and y errors into account, using the York et al. (2001) formulation.

York, D., N. M. Evensen, M. L. Martinez, and J. De Basabe Delgado (2004), Unified equations for the slope, intercept, and standard errors of the best straight line, Am. J. Phys., 72(3), 367, doi:10.1119/1.1632486.

*The figure might suggest a superimposed peak of high ratio values in the mid of 2013.*

We also noticed the mid-2013 peak, but given the large uncertainty, we're uncomfortable making any strong claims about that.

*In Figure 7, the claimed steady rise of the slope during the observation period is hardly recognizable (due to the overlap of data points), perhaps a subdivision in several panels spanning fractions of the whole period would improve the readability.*

We've updated the figure to show the slopes for each year overlaid.

---

## Author Comment (AC3) · 9 Sep 2016

Response to Anonymous Referee #2.

We thank the referee for their valuable comments, which substantially improved the paper.

Referee comments are in *red italics*, our responses are in black text.

*The authors should consider and discuss the statistical significance of the reported trends in observed C1 and C2. The confidence intervals around the annual averages in Table 1, for example, suggest the annual averages across the 3-years shown are not statistically different. On the other hand, assuming the error bars shown in Figure 3 are correct, the 2015 values for C2 emissions seem to be statistically higher than those during 2006-2010. The authors should consider whether the monthly C1 and C2 emission time series in Figure 6 provide an alternative basis to determine the existence of a significant trend (e.g., are the slopes statistically different than zero?).*

We have computed slopes for the monthly emissions. There is no statistically significant trend in the methane emissions during the 2012-2016 period (-9+/-14Gg/yr), and a very slight decrease in acetylene (-0.20+/-0.15Gg/yr). There is a statistically significant increase in the ethane emissions during this period of (1.3+/-0.6Gg/yr). We also looked back at data from two other temporary TCCON stations in the SoCAB (2007-2008 and 2011-2013) for which we can compute methane emissions (but not ethane

or acetylene). Between 2007-2015, there is a (very) slight decrease in methane emissions (-5+/-4 Gg/yr), which is in good agreement with the Wong et al. (2016) estimate of -5+/-4 Gg/yr.

Wong, K. W., T. J. Pongetti, T. Oda, P. Rao, K. R. Gurney, S. Newman, R. M. Duren, C. E. Miller, Y. L. Yung, and S. P. Sander (2016), Monthly trends of methane emissions in Los Angeles from 2011 to 2015 inferred by CLARS-FTS observations, Atmos. Chem. Phys. Discuss., (April), 1–29, doi:10.5194/acp-2016-232.

*Wunch et al (2009) used CO2 instead of CO as the basis to estimate CH4. Also,Wunch et al (2009) pointed out the possible underestimation of CH4 if it was computed from CO emissions, given their differing diurnal profiles (CO emissions primarily influenced by traffic, which was believed to be a stronger daytime source than methane). This new discussion manuscript does not address these issues. The authors should clarify how potential differences in the diurnal profiles of CO, CH4, and C2H6 could affect the emissions estimates calculated with Equation 4.*

Subsequent work has better agreed with the (lower) emissions estimates calculated using CO using aircraft and other remote sensing techniques. A sentence to this effect has been added to the Methods section:

Wunch2009 suggested that using CO instead of CO2 to compute emissions may underestimate the emissions due to different diurnal emissions patterns, but subsequent studies have shown better agreement with the CH4 emissions estimates computed using its relationship with CO [Wennberg2012,Peischl2013,Wong2016].

*The authors should emphasize the importance to their analysis of the changing C2:C1 ratio in pipeline gas. This trend appears to serve as tracer of opportunity, a unique fingerprint that allows attribution of the total observed C1 signal to infrastructure associated with handling, storage, delivery and use of pipeline quality natural gas. This is done indirectly in line 220, but the scientific novelty and utility of the trend deserves greater attention.*

Agreed!

*The manuscript's impact would be improved if the authors could provide a more complete picture about the contribution of specific source types to the observed C1 and C2 trends. Having partitioned the fraction of total methane signal due to pipeline gas (possible due to its increasing ethane content), can the authors further delve into the individual methane and ethane trends and provide a conceptual model that explains the recent trends or patterns in monthly/annual C1 and C2 emissions. [It would seem the C2 emissions might be reducible to a 2-source model (pipeline gas and associated gas/geologic seepage) with appropriate adjustment for vehicle emissions. Similarly, C1 emissions might be reducible to a 3-source model, by adding a generic third term for biogenic C1 sources.] At a minimum, the authors should clearly indicate whether the increasing C2:C1 ratio in pipeline gas is, by itself, sufficient to explain the potentially increasing C2 trend in Fig 3 and 6? Or can the balance of the C2 budget not explained by pipeline gas losses be explained: for example, given likely associated gas compositions, could the local oil/gas production to which Peischl attributed 32 Gg C1 also account for the excess C2 that is not explained by losses of pipeline quality gas? Alternatively, are other causes required?*

There has been a very small decline in the methane emissions over the past 8 years, but no statistically significant change since 2012, when the C2:C1 ratios began increasing. We've created a new figure to show this, that makes use of two TCCON stations that were temporarily in the SoCAB in 2007-2008 and 2011-2013. Overlaid on this plot is the natural gas delivered to SoCAB customers with the y-axis scaled to match the left y-axis if 2% of the natural gas is lost as fugitive emissions.

[Figure]

Assuming a constant methane emission over the 2012-2016 period, the C2:C1 ratio in the pipeline gas is sufficient to explain the increase in ethane since 2012: if we assume CH4 emissions are 413 Gg/yr, and roughly 240 Gg/yr from pipeline natural gas, we would infer C2H6 emissions of 11.6+/-4.4 Gg/yr in 2012-2013, 13.3+/-5.0 Gg/yr in 2013-2014, 15.0+/-5.7 Gg/yr in 2014-2015 using the increasing ethane to methane relationship. Adding this to the Peischl et al. 2010 estimate of C2H6 emissions from local oil and gas, vehicles, and the CARB "other" category (5.4+/-1.0 Gg/yr) results in 17.0+/-4.5 Gg/yr, 18.7+/-5.1 Gg/yr, and 20.4+/-5.7 Gg/yr for 2012-2013, 2013-2014, and 2014-2015, respectively. This falls well within the uncertainties of the ethane emissions estimates from the correlation with CO (19+/-4 Gg/yr 2012-2013; 21.4+/-4 Gg/yr 2013-2014; 23+/-3 Gg/yr 2014-2015).

Attempts to extrapolate this relationship between ethane emissions and C2H6:CH4 ratio back to a regime in which the C2H6:CH4 ratio in the natural gas is zero (to get a sense of the magnitude of C2H6 in the SoCAB in the absence of natural gas C2H6 emissions) is not possible, due to the significant uncertainty on both the monthly C2H6:CH4 slopes and monthly C2H6 emissions. The y-intercept is 7+/-6 Gg/yr, implying that natural gas can explain anywhere from 1/3 to all of the C2H6 in the SoCAB atmosphere.

[Figure]

*Once the C2 budget is determined, and knowing the C2:C1 ratio of pipeline gas, what can the authors say about the trend in C1 emissions due to losses of pipeline quality gas? It would be valuable if the authors could provide an assessment of whether the data indicates that downstream natural gas emissions in the region are changing.*

We can say from our measurements that the methane emissions were roughly constant between 2007-2016 (changing by -5+/-4 Gg/yr), and no statistically significant decline is seen over the 2012-2016 period. According to the EIA, the oil and gas production in the Los Angeles Basin (somewhat larger than the SoCAB) has remained relatively constant over this period (left plot below). Biogenic emissions from the CARB statewide inventory scaled to the SoCAB totaled about 207 Gg in 2007, declining by about 1 Gg/yr due to a 2.8 Gg/yr livestock population change and a partially compensating increase in landfill emissions (right plot below). Thus the small decline we see in atmospheric CH4 emissions (-5+/-4 Gg/yr) might be partially attributable to the decline in biogenics, but the uncertainties are too small to be confident. The downstream natural gas emissions do not appear to be changing significantly.

[Figure]

*It is not clear from the text at line 225 and the reference cited how the authors derive the mass of C1 delivered by SoCalGas to customers within the SoCAB. It is also unclear why sales data going back to 2003 are relevant at this point of the discussion focused on the regional methane budget in 2015 (it would be more relevant – indeed desirable – to show historical gas deliveries in Figure 4). Southern California Gas' annual report (Sempra Energy 2015 Financial Report) reported annual volumes of gas sold in 2013, 2014 and 2015 of 999, 944 and 925 bcf (average 960 bcf, or 17.5 Tg assuming a methane content in gas of 95% ). The authors should explain how they partition the SoCalGas' systemwide sales to isolate the customers solely within the SoCAB. Because not all of the gas sold by SoCalGas is consumed within the SoCAB and may or may not be transported through the SoCAB, the authors should report multiple metrics for the loss of pipeline quality gas that is sold or transported across the basin. One metric would be % of methane delivered that is emitted, and the other is the emissions as a percent of methane throughput in the SoCalGas system. The latter yields a loss rate for pipeline gas of 1.4% of potential throughput (242Gg/17.5Tg). The comparison to Wennberg et al's 2% loss rate should be done with caution, ensuring that the quantities in the numerator and denominator are apples-to-apples between this work and the previous work (it seems*

*the 2% in Wennberg would most appropriately be compared to 1.4%, as calculated above).*

Agree this was unclear. SoCalGas have now published their 2015 delivery numbers, so historical data are no longer necessary. Here is the reworked paragraph:

Since the average total methane emissions in the SoCAB since 2007 have been roughly constant at 413+/-86 Gg yr^{-1}, the ~58\% attributable to the natural gas infrastructure is 240+/-78 Gg yr^{-1}. In 2015, the SoCalGas total throughput was 2559 MMcf day^{-1}, or 18 Tg CH4 total. We remove 3 Tg CH$_4$ from wholesales, and 0.2 Tg CH$_4$ from company use and ``lost and unaccounted for'' (LUAF) gas, giving 14.7 Tg CH4 delivered by SoCalGas. This suggests 1.6+/-0.5% losses as fugitive emissions from the total delivered. (However, only 74\% of the population served by SoCalGas lives in the SoCAB, and thus the fraction of the losses as fugitive emissions would represent a larger fraction of the delivered gas to SoCAB customers [Wennberg2012].)

*Figure 4. The manuscript would be improved if the hydrocarbon production data provided was specific to the SoCAB rather than statewide (these are publicly available from state agencies). Additionally, since hydrocarbon production is only a small contributor to C1 and C2 emissions in the SoCAB, this figure would be much more useful if it presented publicly available activity trends for other chief sources – in particular, I would suggest SoCalGas' natural gas sales and livestock populations. Recent CH4 emissions data or landfills and waste water treatment plants may also be available through the US EPA Greenhouse Gas Reporting program or California state equivalents.*

[Figure]

We have now included Los Angeles Basin production instead of statewide data, which simplifies this analysis somewhat. Furthermore, the left plot above shows the delivered natural gas to the SoCAB (right axis), which is about 11 Tg/year, and the roughly constant CH4 emissions we compute since 2007 (left axis) from our atmospheric measurements. If we assume 2% fugitive emissions, this delivered natural gas represents about 220 Gg/year. The right plot is the emissions from the CARB emissions database for California landfills and wastewater (scaled by SoCAB population relative to California), and enteric fermentation and manure management (scaled to the cattle and calve population in the SoCAB counties relative to California). As described earlier, these biogenics totaled about 207 Gg in 2007 and change only slightly in time. We have therefore added an inventory table that uses our measurements, the 2010 Peischl et al. inventory for biogenics, local oil and gas and vehicles, and included our pipeline natural gas emissions. We compare the sum of the inventory to our atmospheric estimates and the results agree within uncertainties.

*The richest findings seem to derive from the more recent and denser Caltech FTS measurements, with the JPL MkIV FTS data providing corroboration and further insight about historical trends. The manuscript's flow and clarity might be improved with some reorganization of the results and discussion or more explicit delineation of how the two data sets are used to support the conclusions reached.*

We reworked the paper with this in mind.

*The results relating to Aliso Canyon are interesting and important, but are not central to the paper's main findings. I would recommend moving the Aliso Canyon discussion into a separate subsection.*

Done.

*Abstract Line 9. The introduction of "Our methane emissions record" here is confusing since line 4 refers to a record dating back to the 1980s.*

Corrected and reorganized the abstract.

*Abstract Lines 10-15. This wording might be misconstrued to imply that the source of the excess methane is the gas storage facility. In fact the gas storage facility is only mentioned since it is a reliable source of C2:C1 ratios. But the authors have a secondary data source (delivered gas) that yields a statistically indistinguishable trend line in Fig. 5. The authors should revise the language to indicate the comparison is between atmospheric measurements and measured C2:C1 of gas delivered and stored in the region. Additionally, the authors should more explicitly indicate the scope of natural gas infrastructure implicated in the final sentence – to indicate it includes gas delivery infrastructure including pipeline leaks (transmission and distribution), compression and storage facilities, and post-meter losses among others.*

Reworked abstract.

*Line 179. It was unclear how the statement about ethane to acetylene ratios followed from statements about C2:CO and acetylene:CO; please elaborate on the significance.*

Updated text:

There are three main sources of ethane emissions in the SoCAB: vehicle exhaust, the natural gas system, and oil and gas exploration and extraction. Of these sources, only vehicle exhaust is not a significant source of CH4. To distinguish between vehicle exhaust and fossil fuel sources, we use our coincident measurements of carbon monoxide, which tracks sources of incomplete combustion (including mobile sources), and acetylene (C2H2), whose emissions more directly track vehicle exhaust [Kirchstetter1996,Warneke2012,Crounse2009]. The ratio of ethane to carbon monoxide in the SoCAB declined rapidly until the mid-1990s, and then slowly and steadily increased. The ratio of acetylene to carbon monoxide remained relatively constant throughout the time period, and thus the ethane to acetylene ratios follow the same trend as ethane to carbon monoxide. This implies that vehicle emissions are not driving the changes in ethane emissions. This is consistent with the Warneke2012 analysis, which showed an increase in ethane relative to acetylene after 1995, which they attributed to natural gas use and production.

*Line 226. The statement attributing 242 Gg/yr C1 to natural gas infrastructure should be linked back to the prior paragraph's finding that 54% of total excess was due to natural gas (e.g. "242 Gg/yr, equal to 54% of the SoCab total. . .".*

Done.

*Lines 248-255. The specific value used for GWP100 should be stated (e.g., 25, 28, or 34). The choice of 100-yr GWP in this paragraph does not account for the greater short-term climate impacts of CH4. The authors should consider reporting a 20-yr CO2e value in addition to the 100-yr value. The reference to climate impact in the last sentence needs to explicitly distinguish short- and long-term impacts; if only 100-yr GWP comparisons are made, then the sentence should be clarified to refer to "longterm climate impact. . ."*

Done.

*Figure 1. The very rapid rise in C2 mole fraction in the most recent JPL MkIV FTS measurements should be explained (panel 3). Is this trend due to the increased C2:C1 ratio, the Aliso Canyon blowout or both? Should the C2 rise be accompanied by changes in C1?*

Those six high C2 points are on a single day (November 10, 2015), and are due to the Aliso Canyon blowout plume having been advected over the line of sight of the MkIV instrument. The C2 rise is accompanied by a smaller (2.5%) increase in C1, which is difficult to see in the raw CH4 data due to natural variability, but by plotting CH4 versus N2O (see slide 14 of: http://mark4sun.jpl.nasa.gov/report/MkIV_ethene_Toon.pdf), the CH4 increase becomes much clearer.

*Line 141. The word "are" appears twice.*

Removed.

*Figure 1. The black Mauna Loa data points are significantly obscured by the CO and C2 data points.*

Revised figure.

*Fig 3. The error bars are hard to make out and the symbol for the Peischl et al is not evident.*

Figure clarified.

---

## Author Comment (AC4) · 9 Sep 2016

Response to Anonymous Referee #3.

We thank the referee for their valuable comments, which substantially improved the paper.

Referee comments are in *red italics*, our responses are in black text.

*In the paragraph starting on line 187, the authors relate the extraction of petroleum from the SoCAB to the production in the rest of the state. This seems likely to be a valid assumption, but it would be helpful here to provide some additional justification. Would the results of the analysis be substantially different if it is assumed that SoCAB petroleum extraction tracked regional or national trends? Lines 230-235 discuss how non-petroleum sources can close the methane budget. It would be helpful to discuss changes in these sources here to corroborate the conclusion that petroleum accounts for only half of the observed methane increase.*

We have obtained Los Angeles Basin oil and gas production values from the EIA, and have replaced the discussion with the more relevant numbers. This has simplified the interpretation and discussion as it now seems likely that basin oil and gas production can explain the early ethane record from the MkIV measurements.

*The panels on Figures 2 and 3 have "squashed" aspect ratios that make them slightly difficult to read. The bottom panel of Figure 2, for example, compresses much of the data into a small region of the graph.*

We have revised the plots.

*In Figure 3, the presence of four panels in a single figure makes it difficult to see the trends described in the caption. Could some of these panels be merged and their axes modified to make the graphs taller?*

Figure 3 has been reduced to three panels.

*The error bars on the atmospheric ratios in Figure 5 are quite large and imply a large uncertainty in the calculated slope. Indeed, this uncertainty is reflected in the text as well. A visualization of this uncertainty in the figure would be beneficial. Line 219 reports the ratio of slopes as 54 ± 20%, which is thereafter referred to as "about half." However, the large uncertainty in the slope means that the atmospheric increase could be anywhere from not well explained by the changing storage ratios (about 1/3), to very well explained (over 2/3). Do the authors have speculation as to whether the percentage is on the high or low end of this range?*

Error bars have been added to the slope. Since submitting this paper, we have recorded more Caltech measurements, which permitted more robust slopes to be computed. With the new data, we are now able report the mean slope (58%) with a smaller uncertainty (13%).

*Editorial comments In line 244, the slope of the ethane/methane correlation is 4.28 ± 0.07%. This piece of information is in agreement with the storage "ratios exceeding 4%" in line 209. I suggest placing these pieces of information closer together to emphasize this connection, because it provides further evidence that the Aliso Canyon plume was detected.*

We have linked these two numbers better in the revised draft.

*The uncertainties are reported in an inconsistent manner in the text. Line 8 of the abstract contains the quantities 13 ± 4.5 and 25.8 ± 3.9; and line 234 of the text contains the quantity 32 ± 7. Some further discussion of how these different levels of uncertainty for these and other quantities reported in the text were chosen would be helpful.*

Uncertainties have been made more consistent and clearer in the revised draft.

---

## Author Comment (AC5) · 9 Sep 2016

Response to Anonymous Referee #4.

We thank the referee for their valuable comments, which substantially improved the paper.

Referee comments are in *red italics*, our responses are in black text.

*However, some of the conclusions seem to be at odds with another paper currently submitted to ACPD, Wong et al., that concludes that methane emissions in the SoCAB have been decreasing since 2011, albeit with a low confidence interval. Some discussion comparing and contrasting the conclusions of Wong et al. is warranted. For instance, how well does the Caltech FTS represent the entire SoCAB methane emission, compared to the multiple measurement locations described by Wong et al.?*

[Figure]

We went back to TCCON measurements in the SoCAB starting in 2007 to look at longer-term trends in CH4 emissions. From those data, we compute a very small decrease in CH4 emissions of -5+/-4 Gg/yr, which agrees with the Wong et al. -5+/-4 Gg/yr value.

*Line 42, the sampling location of Hopkins et al. and Townsend-Small et al. were heavily skewed toward the western SoCAB. How well do those studies represent emissions to the entire region?*

Added in the introduction that these studies were focused on the western SoCAB.

*Line 110, why do you subtract the daily mean of ethane, CO, and acetylene and not the lowest value?*

We are interested in the daily anomalies, so either method would work. But subtracting off the daily mean has the advantage of producing anomalies with similar values to the Caltech analysis, in which we subtract morning from afternoon values.

*Line 111-113, by aggregating for an entire year, how do you account for this slope not representing the seasonal variability instead of variability due to emissions?*

We remove (to first order) the seasonal variability by computing emissions from diurnal anomalies.

Subtracting the daily means removes the seasonal changes in gas abundances.

*Line 185, is there an earlier reference you could use to support your conclusion that ethane emissions from automobiles would not have accounted for the emissions decline in the late 1980s? The conclusions from the mid-90s on are well supported, but it is unclear they are relevant to the 1980s.*

This is a good point, since many air quality control measures went into place in 1995. However, Kerchstetter et al. note that "... the remote sensors used at the time [of the 1988-1989 study of Bishop and Stedman (1990)] were not capable of measuring VOC or NOx emissions. Thus, the overall effects of oxygenated gasoline or in-use vehicle emissions remain uncertain."

We've added a statement in the revised paper about this:

Thus, emissions from vehicles are unlikely to be either a dominant source of ethane to the SoCAB atmosphere, or responsible for the significant decrease in ethane after 1995. Prior to 1995, there were fewer regulatory controls on air pollution from vehicles, and the exhaust composition is much less well-known [Kirchstetter1996].

Bishop, G. A., and D. H. Stedman (1990), On-road carbon monoxide emission measurement comparisons for the 1988-1989 Colorado oxy-fuels program, Environ. Sci. Technol., 24(6), 843–847, doi:10.1021/es00076a008.

*Line 236, can you confirm with your data that the Aliso Canyon leak did not occur before October 23? There have been some reports of skeptical homeowners questioning that it may have been leaking before this date.*

We see no peaks in our data before October 23, and this is now mentioned in the revised text.

*Line 245, is the ethane emission from Aliso Canyon found by multiplying the 4.28% anomaly by the Conley et al. methane emission of 97.1 Gg? If so, this should be stated more clearly.*

Yes. This has been clarified.

*Line 250, please state which 100-yr global warming potential you used. 25?*

Yes. Clarified in the revised paper.

*Line 262, what is the uncertainty of the 20%? This would help in the comparison with Wong et al.*

This has been removed and replaced by the figure above.

*Line 72, equation 2, a subscripted "dry air" might fit better for the "column dry air", similar to how it is done for the molecular mass?*

Fixed.

*Line 143, Conley et al. state the facility has a capacity of 168 billion cubic feet, and a "working capacity" of 86 billion*

Fixed.

*Line 151-152, Suggest swapping "near the facility" and "from aircraft"*

Done.

*Line 167, change "represents" to "represent"*

Done.

*Line 282, please define "HF"*

Done.

*Figure 4 might look "cleaner" if you used the daily average production for a given month. The variability of the days in a month results in a 3% noise, which is close to the noise between 2003 and 2010.*

This figure has been replaced by one for the Los Angeles Basin, which only has annual values and should look "cleaner".

---

## Author Response (AR2)

October 28, 2016

Dear Dr. Volkamer,

Thank you for your additional comments. Attached is a tracked-changes version of the updated manuscript including two new Appendices (B, C) and three new figures (B1, B2, C1) based on my referee response. I did not create a new Supplementary Information section, but only because I couldn't figure out how to do that within the Copernicus Latex template. I have no preference as to whether the sections are moved to a Supplementary Information section or remain as Appendices during typesetting.

I also wanted to address your concern:

"Also, Fig. 6 of the revised manuscript shows data back to 1985, while the reviewer response shows data exist at earlier times. Is there any reason the data from 1980-1985 are not shown?"

The production data do extend back to 1980, but because our atmospheric measurements only go back to 1985, I did not include the older data in the final figure.

Thank you,

Debra Wunch

[revised manuscript text omitted]